# Antigen persistence and TLR stimulation contribute to induction of a durable HIV-1-specific neutralizing antibody response

Kenta Matsuda [1], Mitra Harrison[1], Eleanor Wettstein [1], Jessica Pederson[1], Alyssa A. Pullano [1], Lyuba Bolkhovitinov[1], Breanna Kim [1], Isabel Steinberg[1], Trevor Griesman[1], Sarah Stuccio[1], Daniel Rogan [1], Andy Patamawenu[1], Tulley Shofner[1], Nathaniel E. Wright [1], Jonathan D. Webber [1], Freya van't Veer [1], Rachel Roenicke[1], Emma Koory[1], Peyton M. Roeder [1], Ellison Ober[1], Benjamin Leach[1], Yaroslav Tsybovsky [2], Tyler Stephens[2], Ivan Del Moral-Sanchez[3], Ilja Bontjer [3], Lori W. McGinnes-Cullen[4], Eric Chu[5], Jason Liang[5], Jonathan L. Torres [6], Ryan N. Lin[6], Andy S. Tran [6], Gabrielle Dziubla[7], Leonid Serebryannyy[7], Sandeep Narpala[7], Bob Lin[7], Mike Castro [7], Gabriel Ozorowski [6], Andrew B. Ward [6], Rogier W. Sanders [3], Peter D. Kwong[7], Javier Guenaga[8], Richard Wyatt[8], Trudy Morrison [4] & Mark Connors [1] ✉

HIV-1 Env glycoprotein (Env) immunogenicity is limited in part by structural instability and extensive glycan shielding and is likely the greatest obstacle to an HIV-1 vaccine. Stabilized Env trimers can elicit serum neutralizing antibodies, but the response is short-lived. Here we use Newcastle Disease Virus-like particle (NDV-VLP) platform to present stabilized versions of HIV-1 Env at high valency and in the context of varied conformational stability, adjuvants, dose, and antigen persistence. Influenza virus hemagglutinin, or SARS-CoV2 Spike-bearing VLPs rapidly induce neutralizing antibodies, in contrast, they were not induced by those bearing Env. A replicating adenovirus type 4 expressing Env rapidly induces autologous neutralizing antibodies. However, durable neutralizing antibodies are induced only when multiple features of a replicating virus infection are combined, with the largest impact from dose and escalating dose. In summary, we show here immunogenicity of HIV-1 Env could be improved by reproducing features of virus infection.

Induction of a durable antibody response capable of neutralizing diverse isolates is among the highest priorities for the development of vaccines for viruses such as HIV-1, SARS-CoV-2, or the influenza virus. In the case of HIV-1, the induction of neutralizing antibodies in humans has been extraordinarily challenging[1]. HIV-1 Envelope (Env), the only neutralizing determinant on HIV-1 virions, has features making it a particularly poor immunogen and difficult target of neutralizing antibodies, including its extraordinary diversity, structural instability, and

extensive shielding by glycans that are sensed as self by the immune system. In many cases, the lack of induction of serum neutralizing antibodies in humans could be attributed to HIV-1 Env not being presented in the appropriate native-like conformation or the use of overly attenuated vector platforms. In recent years, critical advances have been made in using structural biology to produce stabilized native-like Env trimers and other immunogens that open the possibility of inducing trimer-binding neutralizing antibodies with non-replicating

immunogens produced in vitro[2,3]. However, these immunogens do not typically result in rapid development of heterologous neutralizing antibodies[4] and may require many immunizations with adjuvants over months to years, and durability remains an obstacle. Although great progress has been made in producing stable native-like antigens, replicating viral vectors are often more immunogenic, particularly regarding durability[5–8]. This may be due to several factors such as conformation, valency, stimulation of the innate immune response, or persistence of the antigen. However, how these individual factors contribute to immunogenicity remains poorly understood.

Here, we present data on the parameters that might recapitulate features of a replicating virus infection and contribute most to the induction of a neutralizing antibody response. We use stabilizing mutations in Env and a virus-like particle (VLP) system to present Env in a high valency particle. We observe that this display is sufficient to induce neutralizing antibody responses of high magnitude and with durability for antigens such as influenza virus H5 HA and SARS-CoV2 S. However, for HIV-1 Env, this display does not recapitulate responses observed with a replicating vector. To induce responses observed with a replicating vector, we observed that RNA packaging and adjuvants increased response magnitude. However, durability was only improved with high doses and with the use of an escalating dose. These data demonstrate that there are numerous features of a replicating vector that contribute to the rapid development of neutralizing antibodies against HIV-1 that might be used to engineer improved vaccines for HIV-1.

## Results

### Generation of chimeric NDV-VLPs expressing HIV-1 Env, influenza HA, or SARS-CoV2-spike proteins

Env is considered a poor immunogen largely because of the difficulties in generating neutralizing antibodies to HIV-1. However, much of the early work was performed with subunits or non-stabilized forms that might not elicit neutralizing antibodies. Some greater success has followed the development of technologies to stabilize Env[9,10]. Some of these technologies require the removal of the membrane-proximal external region (MPER) to avoid aggregates or express only a portion of the gp120 subunit. However, there are little to no data on head-to-head comparisons of Env to non-HIV-1 surface glycoproteins within a single platform[11,12]. This is particularly true for full-length Env expressed bound to the cell membrane.

To explore the parameters that most contribute to immunogenicity, we used a Newcastle Disease Virus-like particle (NDV-VLP) system that permits the expression of the full-length Env ectodomain on the surface bound to the membrane. This system has been previously shown to be highly immunogenic by inducing higher levels of serum neutralizing antibodies than natural infection in pre-clinical models of RSV infection[13]. We designed VLPs expressing HIV-1 Env, or influenza virus H5 HA or SARS-CoV-2 S as controls (Fig. 1A)[14]. The transmembrane domain (TM) and cytoplasmic tail (CT) of HIV-1 Env were replaced with the corresponding region of NDV fusion protein (F) for incorporation into NDV-VLPs[15]. In this study, the Env sequence of subtype C HIV-1 1086C[16] was used. We generated two HIV-1 Env-NDV chimeric protein constructs with either a covalent disulfide bond and proline stabilization (SOSIP) or native flexible linker trimer-derived (NFL-TD) stabilization to understand how this might impact immunogenicity[3,9,17,18]. For the influenza virus-NDV chimeric construct, we utilized the H5 HA sequence (A/Vietnam 1194/2004)[19]. For the SARS-CoV-2 NDV chimeric construct, only the CT region of SARS-CoV-2 Delta+ S protein was replaced with the corresponding region of NDV F to increase the surface expression of the protein[15]. The expression of these chimeric proteins was confirmed in lysates of transfected cells by western blot (Fig. 1B).

We next produced NDV-VLPs and examined incorporation of chimeric proteins into the particles (Fig. 1C, D). NDV nucleocapsid

protein (NP) was detected in the same preparations with HIV-1 Env, influenza virus HA, or SARS-CoV-2 S proteins, indicating that chimeric proteins are successfully incorporated into the VLPs. When the ratio of Env to NP signal was assessed, stabilized HIV-1 Envs showed higher levels of Env signal, suggesting that the shedding of gp120 is prevented in stabilized Env designs. We further optimized the incorporation of chimeric proteins by co-expressing NDV hemagglutinin-neuraminidase (HN) spike (Supplementary Fig. 1)[14]. The signal of HA protein increased when NDV HN was co-expressed, whereas the signal of HIV-1 Env protein decreased. This result suggested NDV HN protein interacted with the ectodomain of the co-expressed antigens and positively impacted the incorporation of HA protein, but negatively impacted HIV-1 Env and SARS-CoV-2 S proteins. For this reason, the NDV-VLP H5 used in this study contains NDV HN spikes to maximize the amount of chimeric H5 HA incorporation, whereas in the Env and SARS-CoV-2 constructs, it was omitted.

The valency of the expressed antigens on the VLP surface was then assessed by negative-stain electron microscopy (Fig. 1E). Although there was some variability in the size of particles (60–120 nm in diameter), most particles were covered with the spikes with the expected shape, with the exception of VLPs with non-stabilized 1086 C. The computational analysis of images indicated the estimated number of chimeric stabilized spikes was 149 for VLP H5, 104 for VLP CoV2, 120 for VLP SOSIP and 160 for VLP NFL-TD. There were no intact non-stabilized chimeric F/Env spikes detectable on the VLP 1086 C (Fig. 1F), consistent with the western blot result in Fig. 1C. Based upon these densities, the distances between spikes were approximately 233Å for VLP H5, 186Å for VLP CoV2, 158Å for VLP SOSIP and 134Å for VLP NFL-TD. It is notable that the propeller-like shaped HIV-1 Env trimer protein was more readily detected, and the valency was greater on VLP NFL-TD than that of SOSIP stabilized version. Nonetheless, both stabilizing designs successfully prevented the shedding of gp120 subunit and permitted expression at high valency.

The conformational integrity and preservation of broadly neutralizing antibody (bNab) epitopes were confirmed by staining the Env/NDV chimeric proteins expressed on the transfected cell surface with a panel of monoclonal antibodies (Fig. 2A, B, Supplementary Fig. 2A). These included the Env apex (PGT145, PG16), CD4 binding site (CD4bs) (VRC01 and b12), or gp120/41 interface (35O22, 8ANC195, PGT151) and a MPER bNab (10E8). The interface and MPER antibodies were included because they do not bind maximally when native-like Env is membrane-bound due to steric constraints, and binding may suggest some conformational change. In addition, we included antibodies whose epitopes are exposed upon conformational change of Env after CD4 binding; the CD4bs antibody F105, and the V3 antibody 447-52D. Cells infected with an adenovirus type 4 expressing a non-stabilized 1086 C Env (Ad4 FDE3 Env150), a non-enveloped virus that displays the transgene product on the cell surface, were similarly stained as a comparator. Data were normalized to the median fluorescence intensity (MFI) of VRC01 binding, which was used as a measure of the prevalence of Env.

The non-stabilized F/1086 C chimera showed a similar binding pattern to the unmodified full-length 1086 C. The F/1086 C SOSIP construct showed increased binding to some interface antibodies (35O22, 8ANC195, and PGT151), and the MPER antibody (10E8) compared to non-stabilized designs. In addition, the F/1086 C SOSIP construct bound less F105 than non-stabilized designs, consistent with a more native-like conformation. The F/1086 C SOSIP is also bound to the V3 antibody, suggesting some conformational heterogeneity and/or exposure of the V3. The F/1086 C NFL-TD showed no evidence of this conformational change and had markedly reduced binding of antibodies targeting all CD4-inducible epitopes (F105 and 447–52D), all interface epitopes (35O22, 8ANC195, and PGT151), consistent with prior reports[20]. The F/1086 C NFL-TD design did not stain with the 10E8 antibody consistent with the lack of MPER in this design. Although the magnitude of binding differed from the wildtype and non-stabilized F/

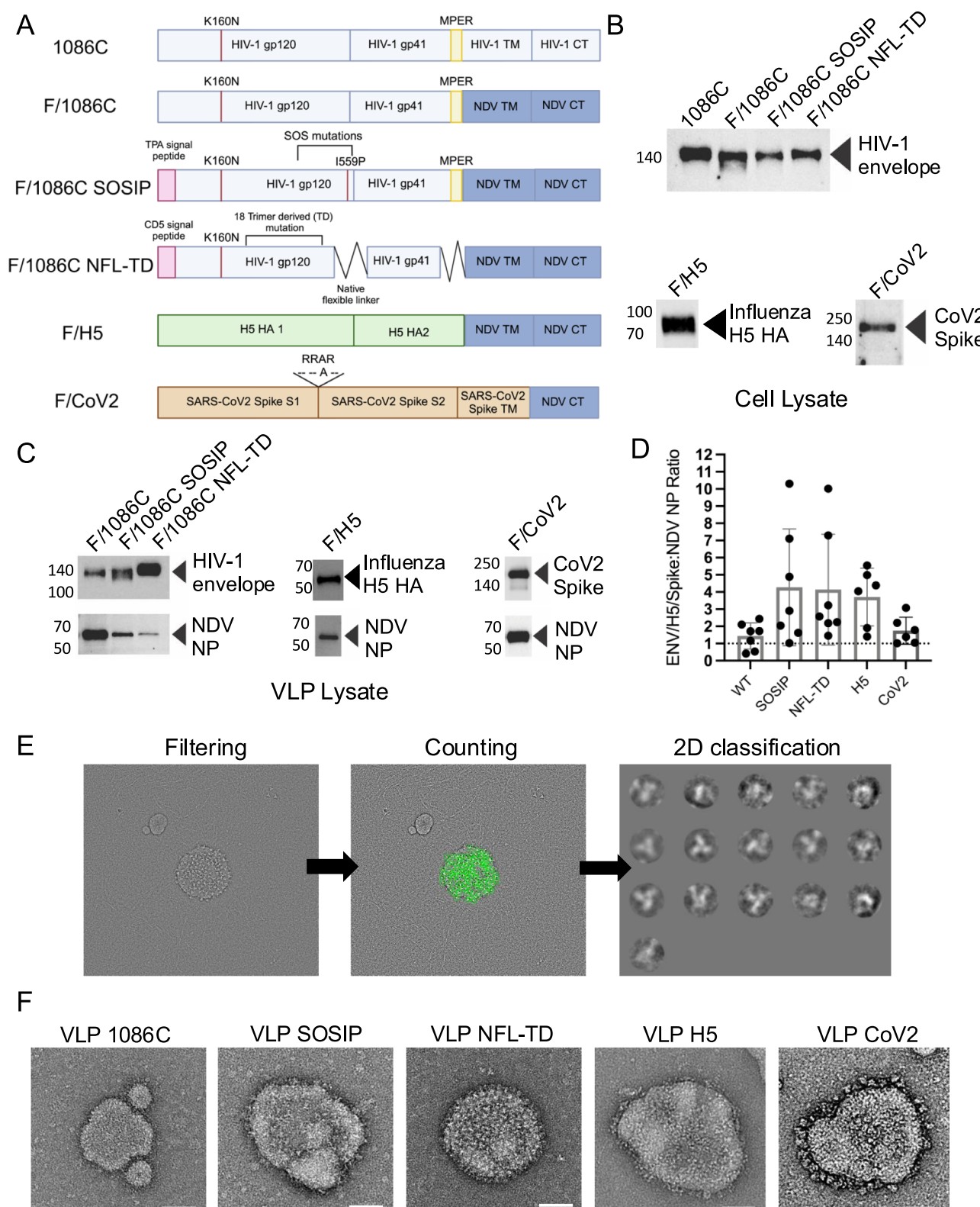

**Fig. 1 | Chimeric NDV VLP production and characterization. A** Schematics of chimeric NDV F expressing HIV-1 Envelope (1086 C, 1086 C SOSIP, 1086 C NFL-TD), Influenza HA or SARS-CoV2 spike proteins designed for this study. **B** Western blots of lysates of A549 cells transfected with the indicated chimeric NDV F chimeric proteins. **C** Detection of indicated chimeric NDV F and NP proteins in purified VLPs by western blot. **D** The ratio of indicated chimeric NDV F protein chimeras to NDV nucleoprotein measured by western blot (1086 C constructs, $n = 7$; H5 and CoV2, $n = 6$). The samples derive from the same experiment, and the blots were processed in parallel. **E** Quantification of protein spikes expressed on the VLP surface performed by negative stain electron microscopy. **F** Representative EM images of VLP variants. The scale bars correspond to 50 nm.

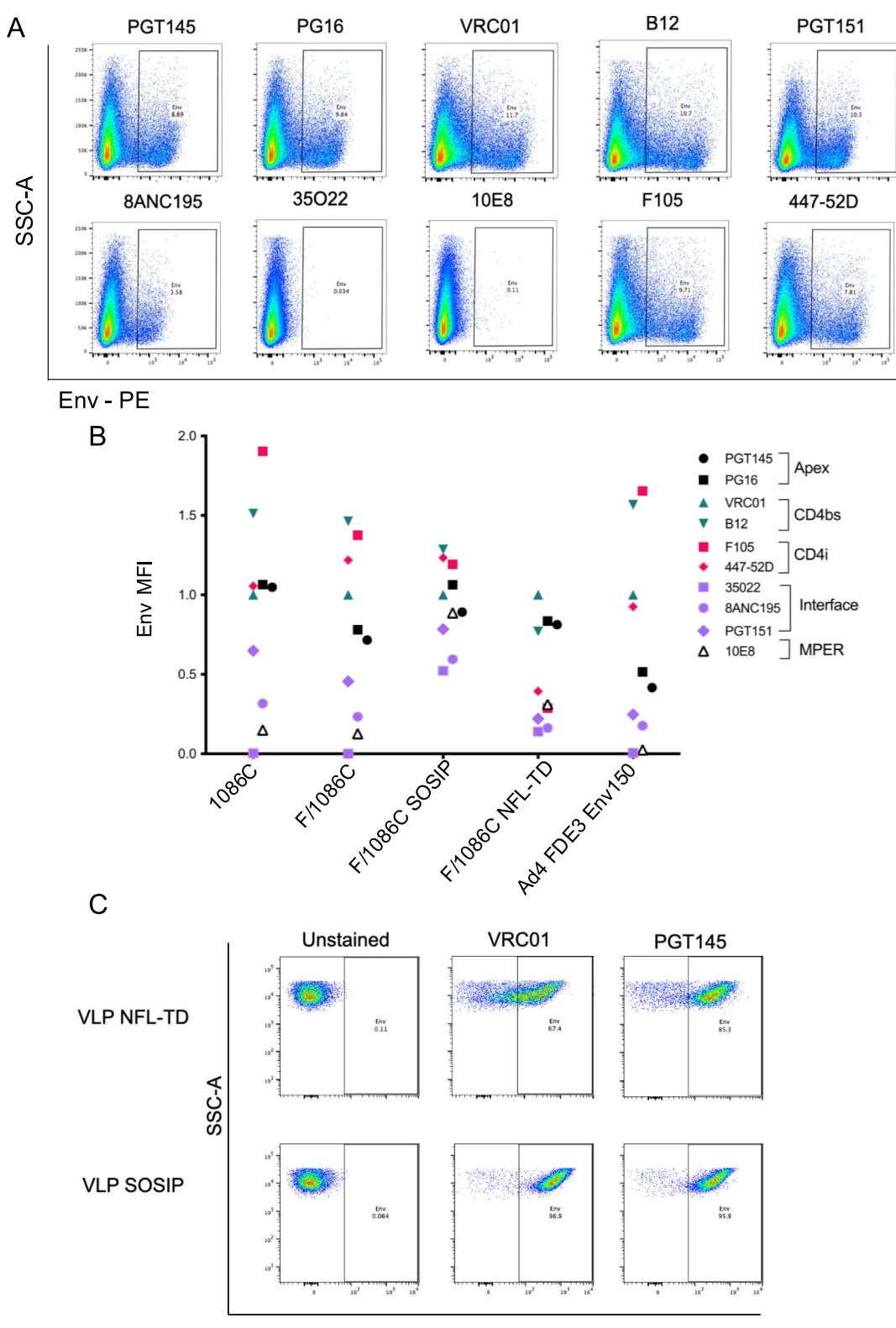

**Fig. 2 | Expression of native-like trimers on producer cells and NDV VLPs.**
**A** Binding of HIV-1 Envelope monoclonal antibodies to chimeric NDV F/1086 C protein on transfected A549 cells by flow cytometry. **B** Summary of HIV-1 Envelope monoclonal antibodies binding to chimeric NDV F/1086 C proteins on transfected, or Ad4 FDE3 Env150 infected A549 cells. MFI is normalized to VRC01. **C** Binding of VRC01 and PGT145 to stabilized 1086 C NFL-TD and SOSIP proteins on chimeric NDV F VLPs by flow virometry.

1086 C, all chimeric constructs successfully formed trimers, indicated by binding of PGT145 and PG16. The level of Env chimeras on the surface of VLPs was further confirmed by flow virometry, which can better detect heterogeneity of incorporation in larger numbers of particles than electron microscopy (Fig. 2C, Supplementary Fig. 2B). Although there was modestly greater heterogeneity in staining of VLP NFL-TD compared to VLP SOSIP, both had overall high levels of staining, indicating good incorporation into VLPs.

## Immunogenicity of VLPs-expressing antigens in high valency

We compared the immunogenicity of VLP H5, VLP CoV2, VLP SOSIP and VLP NFL-TD in groups of 6 New Zealand White (NZW) rabbits, a model of immunogenicity for Env vaccines[21]. Animals were immunized intramuscularly with 150 μg of purified VLPs four times, and blood samples were collected periodically to assess the serum neutralizing antibody activities. The neutralizing activity against influenza virus H5 was detected as early as 4 weeks after the first immunization in all animals immunized with VLP H5 (Fig. 3A). The median neutralization titer peaked at 14 weeks (median $ID_{50} = 5747$; 2 weeks after the 3rd immunization) and decreased to 937.3 $ID_{50}$ by 28 weeks. Similarly, neutralization activity to SARS-CoV-2 variant pseudoviruses was detected in VLP CoV2 immunized animals four weeks after priming (Fig. 3B). The pseudovirus neutralization was high against all tested strains (Peak Delta+ median = 15,156 $ID_{50}$, Wuhan=14266, Omicron B.1 = 13,758, and Omicron B.2 = 11,876) and remained above 1000 at 28 weeks. In contrast, we only detected sporadic neutralizing antibody responses in VLP SOSIP or VLP NFL-TD immunized animals after the 3rd immunization at 14 weeks (Fig. 3C, D). Neutralizing activity was tested against a pseudovirus expressing a heterologous neutralization-sensitive (tier 1) subtype B SF162 that adopts an open conformation[22,23]. It was detected in 3 animals at low levels in VLP SOSIP immunized animals. Most of these activities were diminished by 28 weeks post-immunization. Similarly, neutralizing activity against SF162 and autologous (tier 2) 1086 C was detectable in 3 VLP NFL-TD immunized animals at 14 weeks. To determine the level of immunogenicity that could be achieved with a replicating virus vector expressing the same Env, we also immunized rabbits intramuscularly with $10^{11}$ $TCID_{50}$ of an adenovirus type 4 with a full deletion of the E3 region replaced by the coding sequence for a non-stabilized 1086 C Env with a truncation of the CT (Ad4 FDE3 Env150). Rabbits rapidly developed autologous neutralizing antibodies to 1086 C by 8–12 weeks after this immunization and lower levels of neutralization of SF162 (Fig. 3E, F). This serum neutralizing antibody response showed little, if any, decay over the 28-week study period. These results indicate that Env, even when expressed as a full-length protein bound to the membrane, is poorly immunogenic relative to other viral glycoproteins, and this is not overcome by a native-like conformation and high valency display. They also suggest that some features of the cell surface displayed by the Ad4 viral vector, beyond conformation and valency, can overcome the relatively poor immunogenicity of Env.

## Impact of activation of innate immunity on the magnitude of neutralizing antibody response

One potential mechanism by which presentation of Env by a viral vector might be more immunogenic is through stimulation of innate immunity, enhancing the magnitude of antibody responses[24]. To assess the importance of activation of innate immunity for this Env immunogen, we first co-formulated VLPs with AS01[25–27], a potent stimulator of toll-like receptor (TLR) 4. Rabbits were immunized with 150 μg of VLP SOSIP (Fig. 4A) or VLP NFL-TD (Fig. 4B) formulated with AS01. Serum neutralizing activity against SF162 was detected in 3 animals at 8 weeks in the group immunized with VLP SOSIP. Transient increases in neutralization activity were detected after each boost and peaked at 16 weeks (Median = 302.2 $ID_{50}$). However, these responses fell to undetectable levels in most animals by 24 weeks. Neutralizing

antibodies emerged against SF162 at 4 weeks and 1086 C at 8 weeks in one animal immunized with VLP NFL-TD. More animals developed neutralizing activities over time, and these neutralization titers transiently increased after each boost, although they remained well below 100 $ID_{50}$. Overall, the addition of AS01 did not result in a statistically significant increase in neutralization magnitude for either construct against the SF162 or 1086 C pseudoviruses ($p > 0.05$ for all comparisons) (Supplementary Table 1).

In a viral infection, stimulation of TLRs 7/8 takes place in the endosome of antigen-presenting cells. Some data suggest that much of the AS01 adjuvant may wash through the lymph node and may not be temporally and spatially associated with the vaccine antigen[24]. Therefore, we sought to assess the impact of encapsidating a TLR agonist into the VLP like a virus. RNA40 is a short RNA sequence originally found in the long terminal repeat region of the HIV-1 genome and is known to potently activate TLR7/8 in the endosome of antigen-presenting cells[28]. In NZW rabbits, TLR7 is a pseudogene, although they are known to respond to TLR7/8 agonists[29]. We designed this short RNA fragment such that it would package in VLP particles by flanking it with the NDV leader and trailer sequences (Supplementary Fig. 3A)[30]. The surface Env spike density and conformation of RNA40 incorporated VLPs were like those without RNA40 based upon EM (Supplementary Fig. 3B). Incorporation of RNA in VLP particles was confirmed by qPCR in the presence and absence of RNase during the purification step (Supplementary Fig. 3C). We first assessed the impact of this RNA40 incorporation on the magnitude of neutralizing activity in VLP H5 and VLP-CoV2 immunized rabbits (Fig. 4C, D). Neutralization activity in serum was detected by 4 weeks after immunization and peaked at 14 weeks for VLP H5 and 16 weeks for VLP CoV2, with detectable increases in neutralization after each boost. There was only an increase in neutralizing activities due to RNA40 incorporation in animals immunized with VLP CoV2 and only against the Delta strain (Fig. 4D, C; $p = 0.0321$ for Delta; $p > 0.05$ for all other comparisons). When RNA40 containing Env particles were used, neutralizing activity was observed in two animals immunized with VLP SOSIP RNA40, mostly after the 3rd immunization at 12 weeks (Fig. 4E). However, there were no notable changes in the response rate or magnitude of serum neutralizing activities in the VLP NFL-TD RNA40 immunized group (Fig. 4F, Supplementary Table 1).

To test whether activation of innate responses would induce some breadth of serum neutralizing activity, we tested sera collected at 28 weeks against a panel of 10 HIV-1 strains from subtypes A, B, and C (Fig. 4G). Homologous and heterologous neutralization was low in magnitude. There were only 2 animals with neutralizing activity against more than two strains of HIV-1 in the two groups immunized with VLP SOSIP or VLP NFL-TD formulated with AS01. There was one animal that neutralized 9 strains in the group immunized with VLP SOSIP RNA40. However, no neutralizing activity was detected in the group immunized with VLP NFL-TD RNA40. Overall, there was no statistically significant increase in neutralization magnitude or breadth induced by encapsidation of RNA40 into the Env constructs ($p > 0.05$ for all comparisons). These data suggested that activation of TLR 4 alone or TLR 7/8 may modestly enhance immunogenicity of HIV-1 Env, but this was not sufficient to induce antibody responses comparable to those of VLP H5, VLP CoV2, or replicating Ad4-1086c.

## Impact of dose on the magnitude of neutralizing antibody response

The total amount of antigen presented to the immune system during viral infection is thought to be much greater than that typically achieved by non-replicating vaccines[5,31]. Thus, we assessed the impact of increased dose on the magnitude of neutralizing antibody responses. Rabbits were immunized with 500 μg of VLP SOSIP RNA40 or VLP NFL-TD RNA40 (Fig. 5A, B). Surprisingly, neutralization activity against SF162 was detected at 4 weeks in 4 animals immunized with 500 μg of

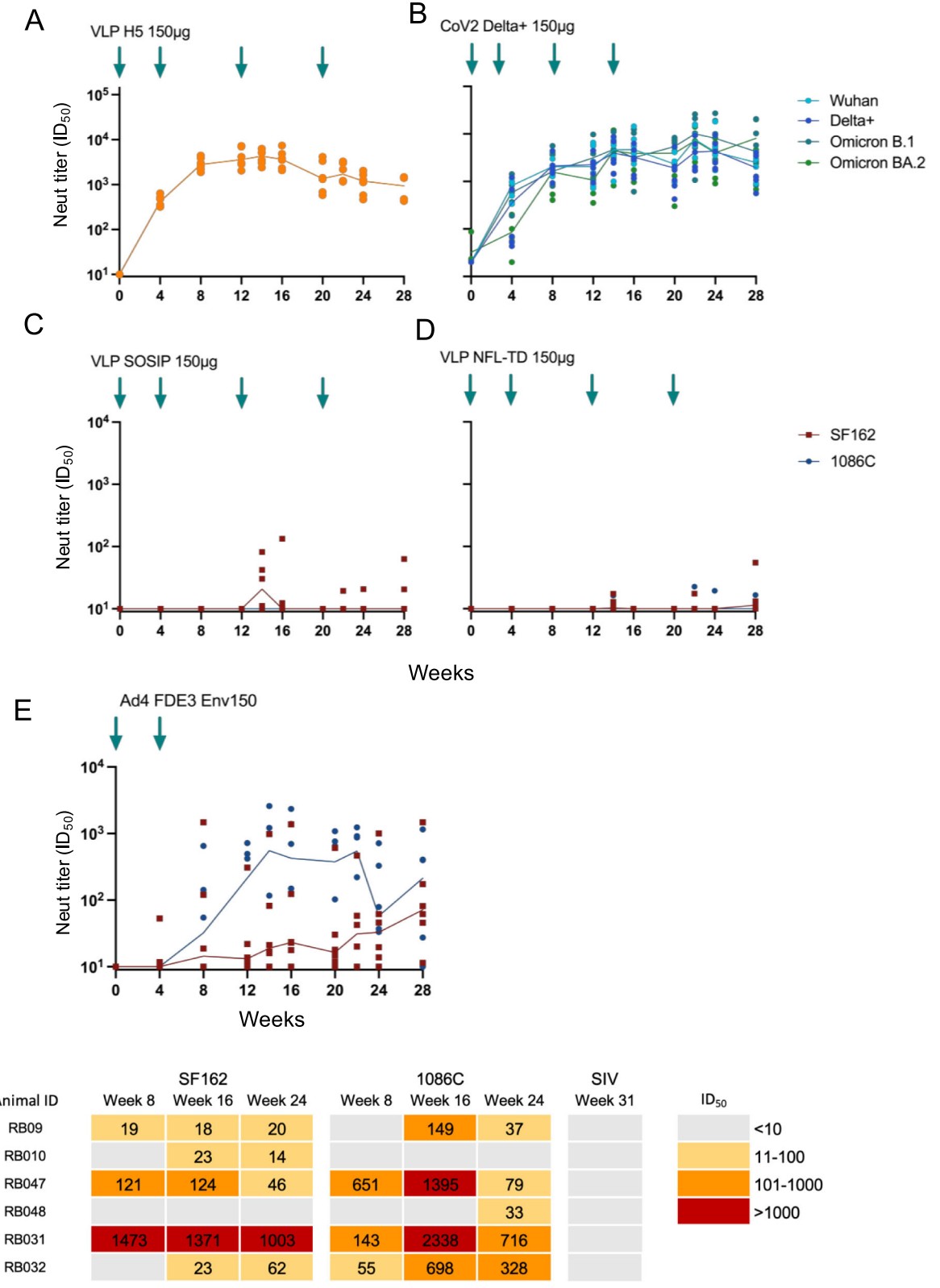

**Fig. 3 | Immunogenicity of high valency chimeric NDV VLP variants in rabbits.**
**A**–**D** Immunogenicity of 150 μg intramuscular (IM) dose administered NDV VLP variant expressing influenza virus HA (**A**), SARS-CoV2 Spike (**B**), SOSIP stabilized HIV-1 envelope protein (**C**), or NFL-TD stabilized HIV-1 envelope protein (**D**).
**E** Immunogenicity of replication-competent Ad4 FDE3 Env150 recombinant ($10^{11}$, IM) expressing non-stabilized 1086 C HIV-1 Envelope protein. Serum HIV-1 neutralizing antibody titers (**C**–**E**) of each rabbit ($n = 6$) are shown in red squares (SF162) and blue dots (1086 C). **F** Serum neutralization breadth ($ID_{50}$) of rabbits ($n = 6$) immunized with Ad4 FDE3 Env150 at the indicated weeks against pseudoviruses of HIV-1 Env SF162, 1086 C and SIV Env mac256. Blue arrows indicate the week of IM-administered immunization. Serum neutralizing antibody titers ($ID_{50}$) of each rabbit ($n = 6$) are shown in dots, and the median value for each time point is connected with a solid line.

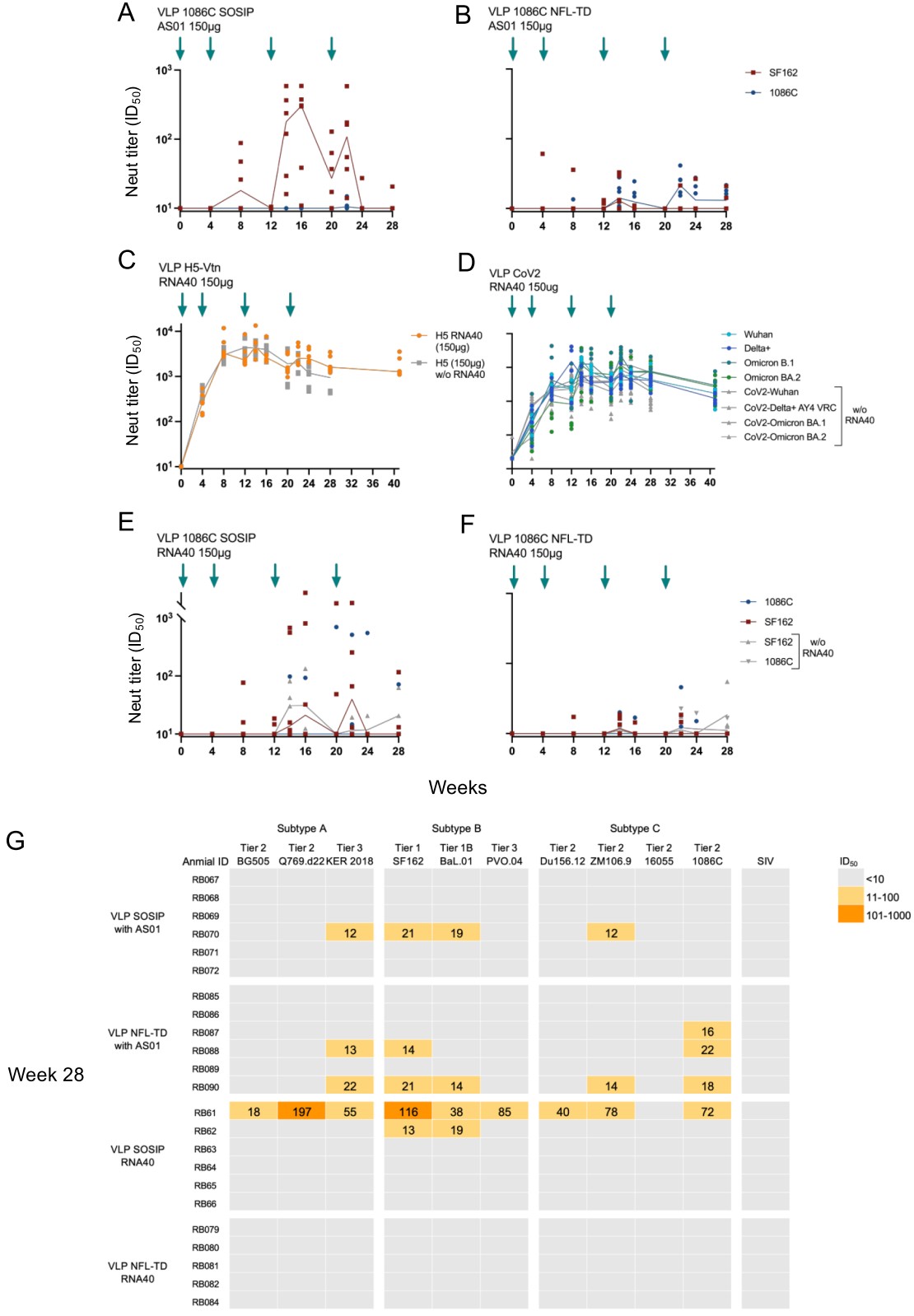

**Fig. 4 | Impact of TLR stimulation on immunogenicity of VLP variants.**
**A–F** Immunogenicity of 150 µg intramuscular (IM) dose of chimeric NDV VLP variant expressing the indicated protein with AS01 adjuvant (**A**, **B**) or TLR agonist encapsidated RNA40 (**C–F**). Animals immunized with constructs without RNA40 are shown in gray. **G** Heatmap of serum neutralizing activity (ID$_{50}$) against a panel of 10 HIV-1 Env pseudoviruses from clades (**A–C**), and SIVmac256. Blue arrows indicate the week of the IM-administered immunization. Serum neutralizing antibody titers (ID$_{50}$) of each rabbit ($n = 6$) are shown in dots, and the median value for each time point is connected with a solid line.

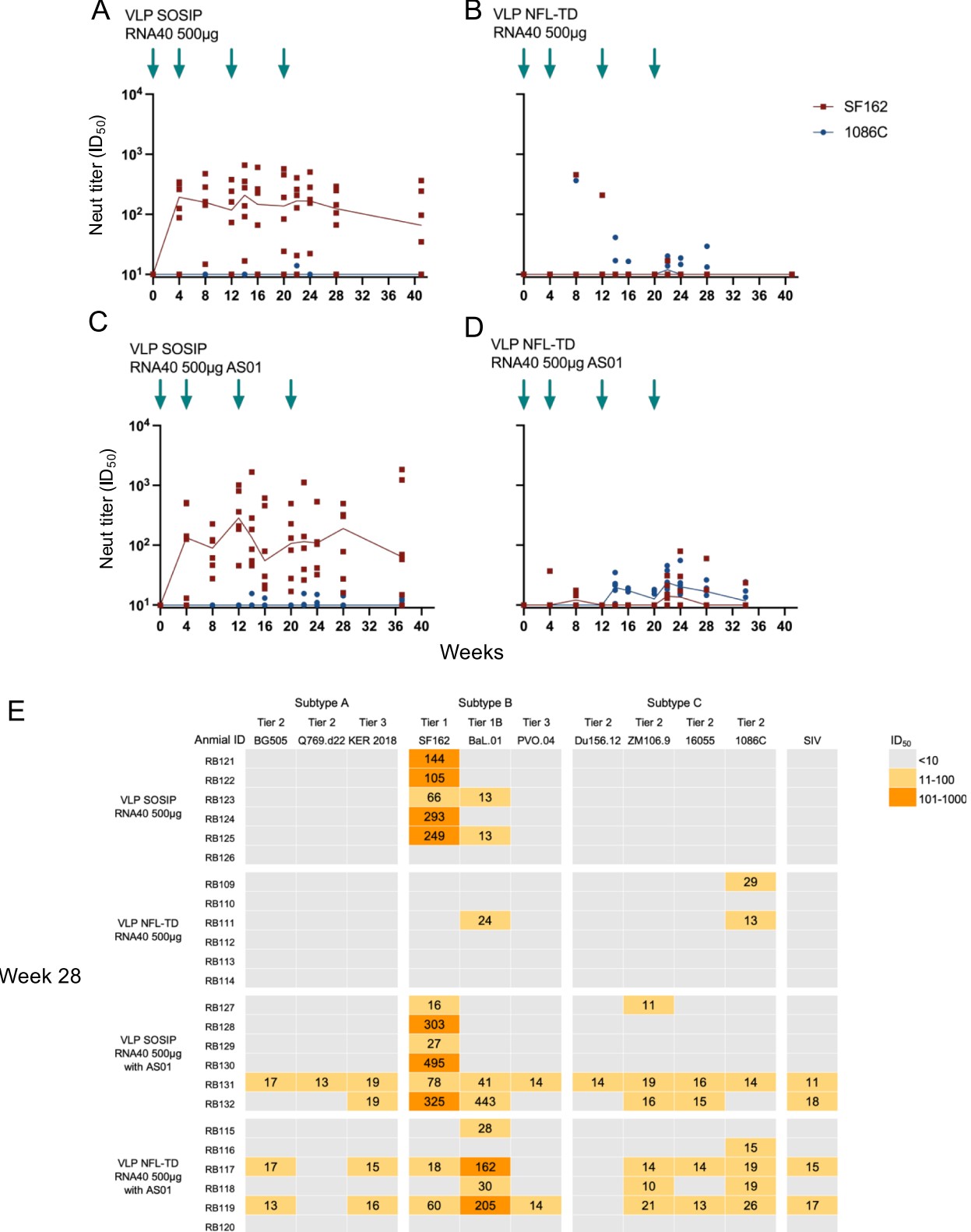

**Fig. 5 | Impact of dose on immunogenicity of VLP variants. A–D** Immunogenicity of TLR agonist RNA40 encapsidated chimeric NDV VLP variants expressing the indicated stabilized HIV-1 envelope protein with or without AS01 adjuvant and IM administered at high doses (500 μg). **E** Heatmap of neutralization activity (ID$_{50}$) in rabbit serum against a panel of 10 HIV-1 Env pseudoviruses from clades (**A–C**), and SIVmac256. Blue arrows indicate the week of IM-administered immunization. Serum neutralizing antibody titers (ID$_{50}$) of each rabbit ($n = 6$) are shown in dots, and the median value for each time point is connected with a solid line.

VLP-1086C SOSIP RNA40. Furthermore, this activity peaked at 14 weeks (Median = 208.2 ID$_{50}$) and did not show a significant decline as seen in the groups immunized with 150 μg of VLP. The serum neutralizing activity at the time of euthanasia (41 weeks) was unchanged from 24 weeks, a considerable improvement in durability compared to lower dose regimens. Although there was an increase in the response rate and the magnitude of neutralizing activity against SF162 this did not achieve statistical significance ($p > 0.05$). No neutralizing antibody against 1086 C was detected in this group of animals, except in one animal at 22 weeks at a very low level. In the VLP NFL-TD RNA40 group, only two animals showed neutralizing activity against 1086C, first detected at 8 weeks. These neutralizing activities were undetectable by 41 weeks post-immunization.

Next, we assessed the impact of activating TLRs 4, 7 and 8 in combination with a high dose (Fig. 5C, D). Some prior data suggest that neutralizing antibody responses increase when adjuvants with these activities are combined[32]. The addition of AS01 alone did not increase the response rate or magnitude of neutralizing activities in high-dose VLP SOSIP RNA40 immunized rabbits (Fig. 5C, Supplementary Table 1) ($p > 0.05$). Although the addition of AS01 only yielded modest improvements in magnitude compared to the 500 μg with RNA40, there were significant increases in neutralization of SF162 in the VLP SOSIP RNA40 group ($p = 0.0360$) and of 1086 C in the VLP NFL group ($p = 0.0329$) when compared to the respective groups immunized with unadjuvanted 150 μg of VLPs (Supplementary Table 1). The neutralizing activities were still detectable at the time of euthanasia (34 weeks). Some increase in breadth of serum neutralizing antibody responses was observed although at a low level (Figs. 4G and 5E). None of the animals immunized with 500 μg of VLP SOSIP RNA40 or VLP NFL-TD RNA40 showed serum neutralizing activities against more than two strains. On the other hand, serum from a total of 5 animals (2 in VLP SOSIP RNA40 and 3 in VLP NFL-TD RNA40 groups) achieved low-level neutralizing activities against more than two strains when the same immunogens were combined with AS01 and given at a high dose. Although some animals developed neutralization of the tier 1B viruses SF162 or BaL.01, in most cases, this broadening of the response was associated with low-level neutralization of the SIV pseudovirus, which is a specificity control. These results suggested that the total amount of the antigen presented to the immune system has a significant impact on the magnitude as well as durability of the antibody response, but repeated immunization at high doses with adjuvant can induce some nonspecific neutralization.

## Impact of dose escalation on the magnitude of neutralizing antibody response

Another characteristic of a viral infection is in vivo amplification of the antigen. It has been reported that immunization in a dose-escalating manner increases the immunogenicity of soluble trimer-based vaccines[33]. This is thought to work by early immunogen-specific binding antibodies causing improved lymph node retention of antigen and germinal center formation. To test the impact of dose escalation, we split 500 μg of VLP SOSIP RNA40 or VLP NFL-TD RNA40 formulated with AS01 into 7 doses and immunized rabbits in a dose-escalating manner over two weeks. Most VLP SOSIP RNA40 immunized rabbits developed serum neutralizing activities against SF162 at 4 weeks, and continued to increase over time, peaking at 24 weeks (after the 4th immunization) (Fig. 6A). Neutralizing antibody responses against 1086 C were detected in most animals by 22 weeks (Median = 19.6 ID$_{50}$), although most of these responses were below 50 ID$_{50}$. The increase in neutralization provided by dose escalation was statistically significant for 1086 C ($p = 0.0038$). An effect of dose escalation was also observed in the animals immunized with VLP NFL-TD RNA40 (Fig. 6B). There was a higher response rate against both SF162 and 1086 C, peaking at 24 weeks (Median; SF162 = 32.6 ID$_{50}$, 1086 C = 70.8 ID$_{50}$). However, the serum neutralizing antibody response against

these viruses was not durable and waned to below the level of detection by week 41 in all but 1 animal and did not achieve statistical significance ($p > 0.05$, for SF162 and 1086 C).

There was also an increase in the number of animals showing breadth in neutralizing activity in both groups (Fig. 6C, Supplementary Table 2). Five out of six animals immunized with an escalating dose of VLP SOSIP RNA40 showed neutralization against more than 2 strains (no dose escalation vs escalation; $p = 0.0289$). Three of these animals were able to neutralize all tested strains, albeit at low titers. Similarly, 3 of 6 animals immunized with VLP NFL-TD RNA40 were able to neutralize all the tested strains, including some difficult to neutralize tier 2 and 3 strains. Although the response magnitude for most animals was low, the result is surprising given that this was induced with a single strain immunogen and did not require heterologous boosting. However, this low-level response again extended to the SIV control pseudovirus, indicating a lack of specificity. Prior work suggests that neutralization of Tier 1 viruses such as SF162 might be dominated by antibodies with specificity for V3[34]. To evaluate this possibility, we performed competition experiments with V3 peptide (Supplementary Fig. 4). The neutralizing activity against SF162 induced by VLP SOSIP was competed ($p = 0.0146$), suggesting that this activity was V3 mediated, likely induced by V3 loop exposure within the immunogen, consistent with the surface staining in Fig. 2B.

We next sought to further understand the mediators of the low magnitude, broad, but nonspecific responses observed in the dose escalation experiments. We first purified immunoglobulins from the serum of animals in Fig. 6C to determine if the response was mediated by the humoral immune response or by some other non-specific factor in the sera. Overall, the level of neutralization in sera tracked with the IC$_{50}$ observed after purification (Supplementary Fig. 5A, B). This purified immunoglobulin also mediated neutralization of pseudoviruses bearing murine leukemia virus Env and vesicular stomatitis virus G proteins (Supplementary Fig. 5C, D). These results suggested that the broad, low-level neutralization we observed was immunoglobulin-mediated, but not Env-specific.

The induction of binding antibodies was also examined in sera. Binding was measured on week 28 sera using a Meso Scale Discovery assay against 3 trimers; the heterologous (clade A) BG505 that has additional disulfide (DS) stabilizing mutations (201 C, 433 C) (BG505 DS-SOSIP)[35], and the homologous 1086 C DS-SOSIP and 1086 C NFL-TD (Fig. 7A). Binding antibodies to the BG505 DS-SOSIP trimer were primarily induced by the 1086 C SOSIP immunogen and largely paralleled the neutralization results against SF162 shown in Figs. 3–6. The previous V3 competition result suggests that this binding might be largely mediated by V3-targeting antibodies. However, high levels of binding antibodies were induced to both the DS-SOSIP stabilized and NFL-TD trimers by the NFL-TD immunogen but were not induced by SOSIP immunogens (Fig. 7A). Taken together with the neutralization data, the VLP regimens using the SOSIP stabilized 1086 C induced neutralizing antibodies that target V3 that may be responsible for SF162 neutralization, in addition to other specificities responsible for the heterologous Tier 2 virus neutralization. The VLP regimens that were stabilized with NFL-TD induced high levels of trimer-binding antibodies but relatively low levels of serum neutralizing activity.

We also attempted to map the specificities of binding antibodies that might mediate the broad, low-level responses using electron microscopy polyclonal epitope mapping (EMPEM). Sera from animals at day 41 were used to bind the 1086 C NFL, BG505 SOSIP[2] or the PVO.04 NFL[23,36] trimers. Binding of antibody binding fragments (Fabs) from the sera of most animals with low-level neutralization was not detected by EMPEM (Supplementary Fig. 6A). Binding was only detected in sera from rabbit RB142, which had the greatest breadth and magnitude of serum neutralization. In this NFL-TD immunized rabbit, binding was only observed to 1086 C NFL, and not to BG505 SOSIP or PVO.04 trimers (Fig. 7B, Supplementary Fig. 6A). Most trimers bound

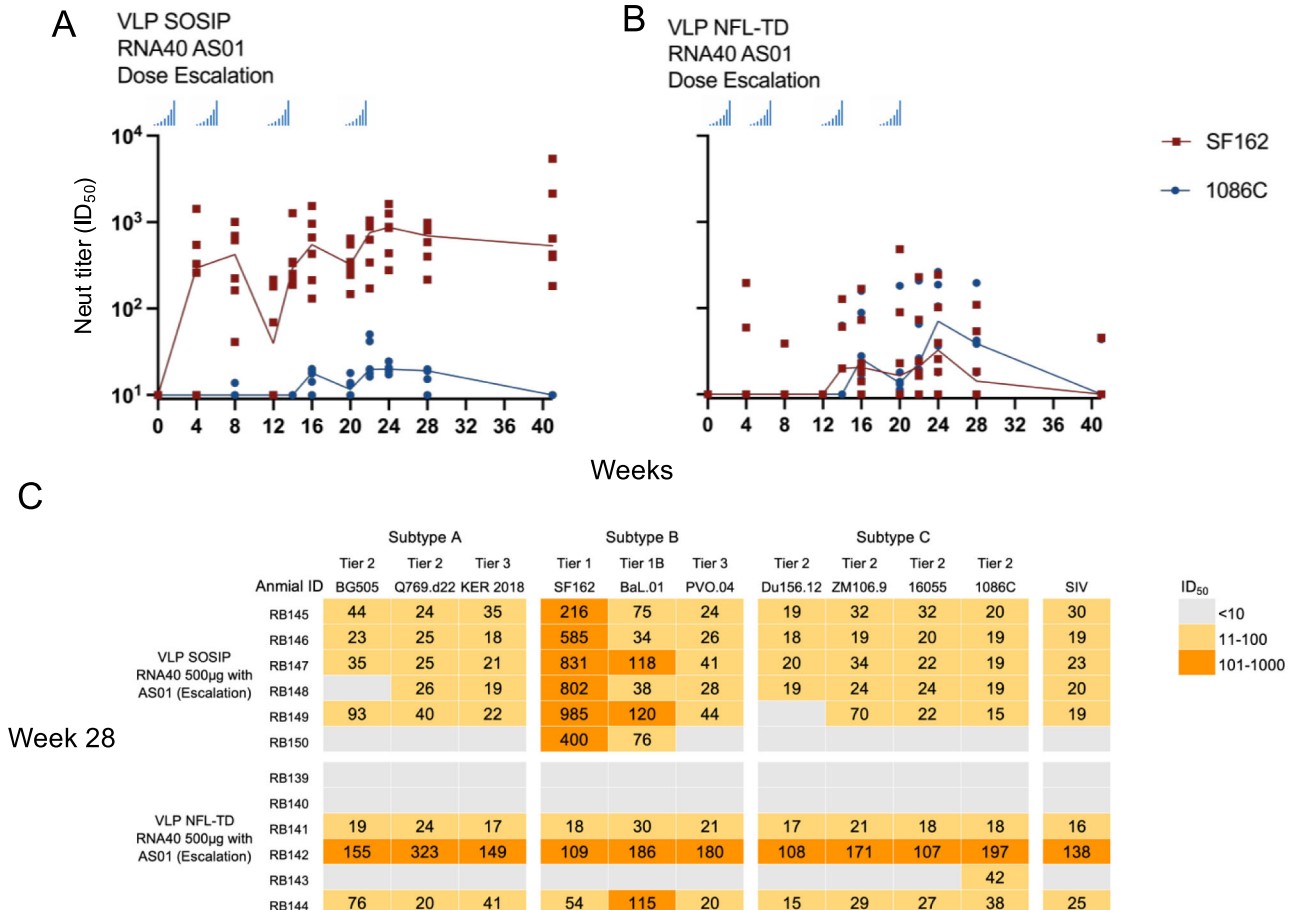

**Fig. 6 | Impact of dose escalation on immunogenicity of VLP variants.**
**A**, **B** Immunogenicity of TLR agonist RNA40 encapsided chimeric NDV VLP variants expressing the indicated HIV-1 envelope formulated with AS01 adjuvant given at an escalating dose of 500 µg. The immunogen was split into 7 IM immunizations in a dose-escalation manner. Four separate weeks of dose-escalation IM immunizations are indicated by the blue histogram. Serum neutralizing antibody titers of each rabbit are shown in dots, and the median value for each time point is connected with a solid line. **C** Heatmap of rabbit serum (Week 28) neutralization activity (ID50) against a panel of 10 HIV-1 Env pseudoviruses from clades (**A**–**C**) and SIVmac256.

up to 3 polyclonal antibody binding fragments (Fabs) against the V3/V1 region, centered on N301 (HxB2 numbering). It is notable that 1086 C lacks N295 and has a potential N glycosylation site at N334, potentially reducing steric hindrance in this area and increasing the likelihood of detection of antibody binding. In addition, the pFab-bound 1086 C NFL did adopt a more open conformation compared to other trimers (Supplementary Fig. 6B). This raises the possibility that these antibodies require a more open conformation and were not captured on the BG505 SOSIP or PVO.04 trimers that adopt a more closed conformation and are less likely to expose these epitopes.

## Discussion

In this study, we explored the factors that contribute to immunogenicity that might be exploited to engineer better immunogens. The NDV-VLP system permitted variation of many of the features of a live virus infection thought to contribute to immunogenicity, including the surface glycoprotein and its conformation, spike density, adjuvant, packaging of RNA as a TLR agonist, total and escalating doses. We observed that HIV-1 Env is markedly less immunogenic than H5 or SARS-CoV2 Spike in the NDV VLP system, and this observation adds to the notion that HIV-1 is exceptionally poorly immunogenic[11,12]. This lower immunogenicity was not attributable to the number of glycoprotein spikes, given that the valency of the Env VLPs was intermediate between VLP H5 and VLP CoV2. Although overall glycoprotein spacing was higher than the 50–100 Å spacing thought

to be important for HPV VLP vaccine immunogenicity[37], the spacing was greater for VLP H5 and VLP CoV2 compared to Env VLPs, suggesting this was not a cause of the differences in immunogenicity. We also observed that there was some modest improvement in the induction of neutralizing antibodies with the incorporation of a TLR 4-stimulating adjuvant and the packaging of a TLR 7/8 agonist RNA. However, by far the largest impact on the durability of the neutralizing antibody response was observed with large increases in total dose or use of an escalating dose scheme. It is important to note that poor immunogenicity was specific to HIV-1 Env. The inclusion of adjuvants or very high doses, were not required to induce magnitude or durability to H5, S, or in prior work, RSV F[13]. This suggests that poor immunogenicity is not simply a product of the NDV system. Rather, this is possibly a result of the closed conformation of Env primary isolates, closed conformation of highly stabilized Envs, their extensive coating with self-glycans, or other factors.

At first examination, it may seem somewhat surprising that dose would play such a prominent role in the induction of durability of the neutralizing antibody response. However, there are some examples in human clinical trials that are consistent with these results. In a prior trial, we observed that when Ad4 H5-Vtn was given by the oral route, the serum neutralizing antibody response waned by 6 months[6]. However, when the same vaccine was given by the intranasal route, where there is greater replication, neutralizing antibodies were unchanged from their peak at 3–5 years after immunization. In another example,

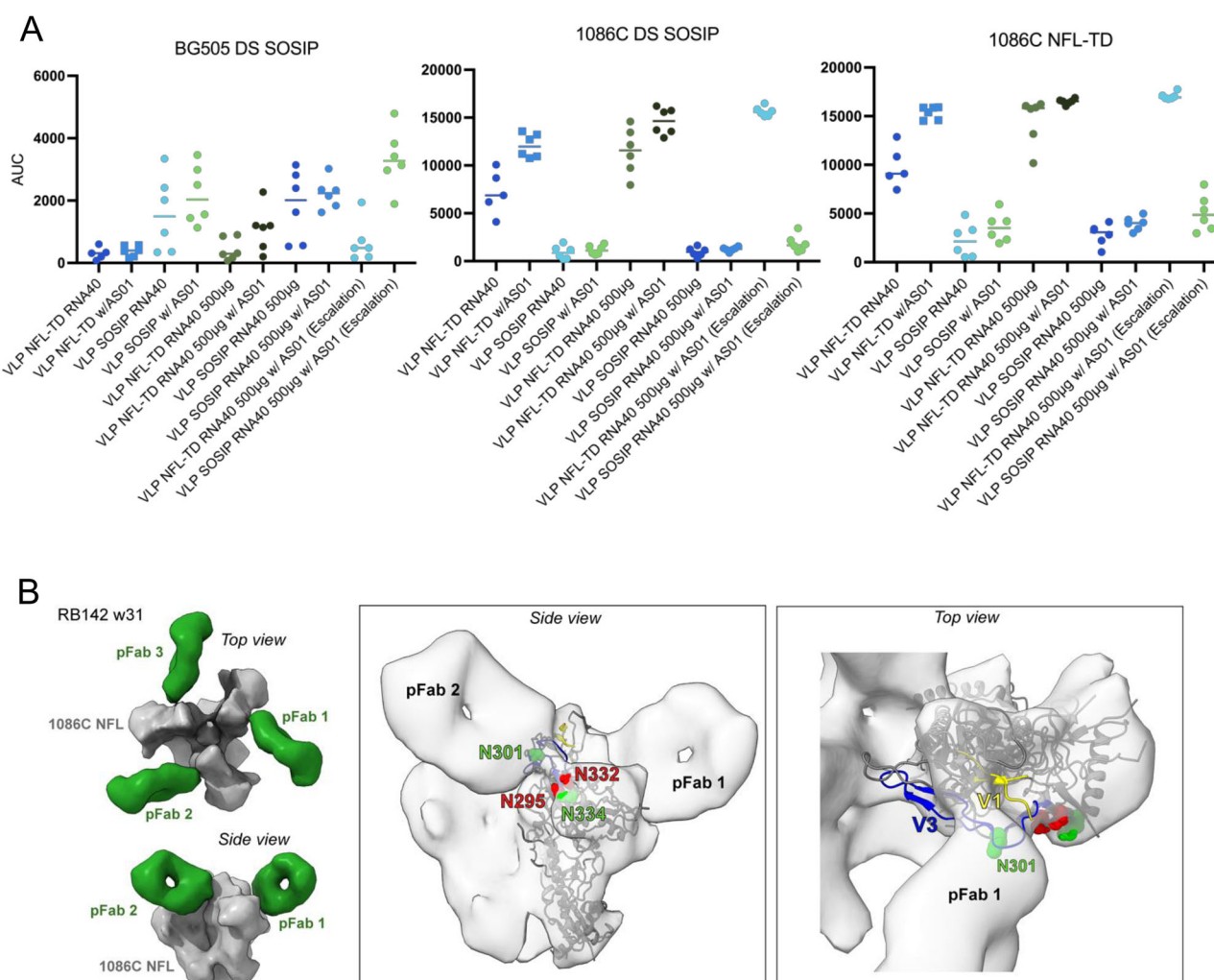

**Fig. 7 | Binding assay and EMPEM. A** Sera from week 28 from rabbits ($n = 6$) immunized with the indicated vaccine regimen was assayed against the indicated stabilized trimer protein. The area under the curve (AUC) of serial dilutions is displayed. **B** Negative-stain EMPEM 3D reconstruction of polyclonal antibody fragment antigen-binding (pFab) from rabbit 142 (RB142) at week 31 in complex with 1086 C NFL trimer (left). The map is segmented and colored by component.

Docking of an Env gp140 protomer model into the map reveals that the polyclonal antibodies bind near the V3 and V1 regions of gp120, near N301 (middle and right panels). Relevant potential N-linked glycosylation sites in 1086 C NFL are labeled in green, and those commonly present in other HIV-1 genotypes but not in 1086 C are labeled in red.

the half-life of antibodies to vaccinia virus is measured in hundreds of years[5]. However, serum responses to the modified vaccinia Ankara vaccine, which does not replicate in human cells, last less than 1 year even after 2 doses[38]. Even with non-replicating vaccines, such as the inactivated Hepatitis A vaccine, large increases in dose can extend the durability of the serum antibody response to more than 7 years[39]. In addition, there was a recent demonstration that the use of self-amplifying RNA vaccine for SARS-CoV-2 can dramatically increase the durability of serum neutralizing antibodies compared to conventional mRNA vaccines[40]. It should be noted that the doses used here for induction of a neutralizing antibody response to Env when scaled to a 70 kg human, are ~18 mg per dose. This compares to only 100 mcg of Env trimers in recent clinical trials[41] and 40–50 mcg of total protein for the HPV VLP vaccine. Such doses would likely be prohibitive for widespread use in humans. Not only was the total dose an important factor in driving durable antibody responses, but also the use of an escalating dose. At present, the mechanism of the effect of higher doses in inducing long-lived plasma cells is incompletely understood. These might be related to greater persistence of antigen in draining lymph node and driving of germinal centers, or presentation of antigen by alternative modes or cells[33].

The finding that the addition of adjuvants increased the magnitude but did not result in improvements in durability is consistent with several published reports. For example, formulation of chimeric influenza split vaccines with AS01 dramatically increases the magnitude of the antibody response but does not dramatically improve the poor durability of the response[42]. A similar increase in magnitude but lack of improvement in durability was observed with split H5 influenza vaccines formulated with AS03[43]. This is in contrast to the pattern of durability of responses we observed to a replication-competent Ad4-H5-Vtn given intranasally, where a single dose stimulated B cell expansions, somatic hypermutation and affinity maturation that continued for 6 months to one year[44]. When vaccinees returned for boosting at 3–5 years, the anti-H5 serum neutralizing titers were unchanged from their peak[6]. The short-lived nature of the serum antibody response to SARS-CoV2 S induced by currently licensed mRNA vaccines, and prior experience with mRNA vaccines for H7, H10[45] and Rabies gp150[46], and some protein-based vaccines[43] underscore the need to better understand the parameters that guide antibody durability.

It was also surprising that high valency display in a native-like conformation, even with the addition of adjuvants, only induced

very modest titers of neutralizing antibodies and very limited durability. High valency display is thought to improve the stimulation of naive B cells by overcoming a low-affinity interaction with a highly avid one that engages multiple B cell receptors per cell[47]. This display may also enhance antigen uptake by B cells and presentation to CD4[+] T follicular helper cells ($T_{FH}$)[48]. Induction of CD4 + $T_{FH}$ cells is thought to be a critical determinant of the induction of long-lived plasma cells in the bone marrow and thus antibody durability. In large part, this is thought to be the mechanism for the success of the human papillomavirus VLP vaccine that induces neutralizing antibodies that remain above the protective threshold for the life of the vaccinee[5,47]. Critical to this activity is the rigidity of the immunogen and spacing of surface spikes. However, high valency display of HIV-1 Env in a native-like conformation appears insufficient, under the current experimental conditions, to induce a durable neutralizing antibody response.

There are some important limitations to this study. First, we were unable to fully understand the mechanism of the low-level non-HIV-1-specific neutralization that we observed in highly immunized animals, although it appears immunoglobulin-mediated. It is highly unlikely that such low-level responses, if achieved in humans, would meaningfully impact HIV acquisition[49]. In addition, we should underscore that even with high valency display, adjuvants and an escalating high dose, we did not achieve the magnitude of autologous neutralization that we observed with a replicating Ad4-1086c. Of the available vaccine platforms for presenting viral glycoproteins to the immune system, replicating vectors have several advantages over most non-replicating vaccines. They can express viral surface glycoproteins on the host cell at high valency, over a prolonged period, and in the appropriate conformation and glycosylation state. Antibodies induced by the host cell, which produce glycoproteins, in contrast to those produced in cell lines or eggs, may better target virions during natural infection. Replicating vectors may also directly or indirectly stimulate B cell proliferation and differentiation through nucleic acid stimulation of toll-like receptors in B cells or antigen-presenting cells and induce pro-inflammatory cytokines. Some combination of these factors likely leads to the extraordinary durability of the neutralizing antibody response to replicating vectors[5–7]. Although replicating vectors offer numerous advantages, the level of immunogenicity is often modulated by the level of replication of the vector, transgene expression, pre-existing immunity, and route of administration. In addition, development can be complicated by safety or transmission concerns. One path forward is to further develop replicating vectors and address these issues, and in parallel, address the reasons for the differences in immunogenicity between replicating and non-replicating approaches. In theory, it should be possible to achieve the magnitude and durability of neutralizing antibodies induced by a live virus with a non-replicating immunogen. Additional modifications to the stabilization strategy or glycan shielding could further improve Env immunogenicity for both approaches. It is also possible that there are other factors that contribute to immunogenicity that were not explored here, such as tissue injury or T follicular help, that might be pursued in further work. The parameters that approximate a replicating virus infection found here may therefore serve as an important starting point for such work. However, the rapid generation of 1086-specific neutralizing antibodies by the Ad4 vector suggests that poor immunogenicity is not due to factors solely intrinsic to Env, such as glycosylation, or lack of engagement of the naive B cell repertoire. Rapid induction of autologous neutralization by an Ad4 recombinant suggests that there are features of a live virus infection that can overcome the poor immunogenicity of HIV-1 Env, and if better understood, might further the development of vaccines, replicating or non-replicating, that might accelerate the induction of neutralizing antibodies.

## Methods

### Ethics statement

All animal experiments were conducted according to the animal study proposal (ASP LIR21/LIRID9) approved by National Institutes of Health (NIH), National Institute of Allergy and Infectious Diseases (NIAID), Animal Care and Use Committee (ACUC) that meets all federal requirements, as defined in the Animal Welfare Act (AWA), the Public Health Service Policy (PHS), and the Humane Care and Use of Laboratory Animals in AALAC accredited facilities. This study used 6–8-week-old female NZW rabbits (Charles Rivers Laboratories, Wilmington, MA, USA) that were co-housed, all of which were processed for terminal bleed collections under general anesthesia and euthanized by exsanguination as approved by the AVMA (American Veterinary Medical Association) and adopted by NIH-NIAID ACUC.

### Construct design

The Env sequence for VLP and adenovirus constructs was derived from the clade C 1086 isolate (name: 1086-B2 C, GenBank accession number: FJ444395, from Malawi 2004)[16]. The original 1086 C was modified to introduce a K160N mutation to permit binding of antibodies specific for the apex (HxB2 numbering). The SOSIP version was generated by adding the following changes: a TPA signal sequence (MDAMKRGLCCVLLLCGAVFVSPSQEIHARFRRGAR), A501C and T605C (gp120-gp41$_{ECTO}$ disulfide bond), I559P in gp41$_{ECTO}$ (trimer-stabilizing), H66R and T316W (trimer-stabilizing), Q543N in gp41$_{ECTO}$ (improved trimerization) and REKR to RRRRRR (R6) in gp120 (furin cleavage enhancement)[2,50,51] and a stop codon after residue 704. A second intermolecular disulfide bond was added by introducing mutations A73C in gp120 and A561C in gp41[52]. Eight BG505 Trimer Derived (TD) mutations were also introduced (E47D, K49E, V65K, E106T, E429R, R432Q and E500R; the sequence already contains a L at position 165)[10,53] and MD39 mutations (T106E, R304V, A319Y, P363Q, F519S, L568D, V570H and R585H; the sequence already contains an I at position 271 and a L at position 288)[54] resulted in 1086c SOSIP.v8.2 gp145.

To generate an NFL trimer design, the furin cleavage site REKR (HIV-1 Env residues 508–511) was replaced by a flexible linker (GGGGSGGGGS) to covalently link the gp120 and gp41 Env subunits[20]. The natural HIV-1 Env leader sequence was replaced by the CD5 leader sequence to increase expression. The following HIV-1 Env TD substitutions were made to generate highly stable and homogeneous NFL trimers: E47D, K49E, V65K, E106T, E429R, R432Q, E500R; helix-destabilizing gp41 mutations, I559P, L568G, N636G; V3 and Fusion peptide stabilizing mutations, N302Y, T320M, F519R, L520R and V513Y. To further enhance sensitivity to the V2-apex antibodies, the K166R and H170Q mutations were also introduced[3,10]. Finally, a second linker GGGGS was incorporated to replace the MPER, residues 664–683, to covalently link the 1086c NFL to the NDV TM. The Ad4 FDE3 Env150 was constructed as previously described except that a stop codon was introduced at position 732 to enhance surface expression, and the REKR furin cleavage site was restored to improve antigenicity[55]. The 1086c Env also included the K160N mutation.

### Plasmids and VLP production

Plasmids for the production of NDV VLP were generated as previously described[14]. Briefly, B1 strain of NDV cDNA sequences encoding NP, M and HN were subcloned into the mammalian expression vector pCAGGS to generate plasmids for co-transfection in the construction of VLPs. Chimeric constructs were constructed by combining protein coding regions of either Influenza H5 HA, HIV-1 Env 1086 C, HIV-1 Env 1086 C SOSIP, HIV-1 Env 1086 C NFL-TD or SARS-CoV2 S Delta+ with the TM and CT from NDV fusion (F) protein. F/H5, F/1086 C, F/1086 C SOSIP, F/1086 C NFL-TD and F/CoV2 Delta+ plasmids were generated by synthesizing the chimeric codon optimized F protein sequence containing ectodomain sequence from Influenza H5 HA (A/Vietnam/

1203/2004; GenBank accession EF541402), HIV-1 1086 C Env, SOSIP stabilized HIV-1 1086 C Env, or NFL-TD stabilized HIV-1 1086 C Env respectively (Genscript, Piscataway, NJ, USA). The SARS-CoV-2 Delta+ full-length spike protein sequence (GenBank accession number OK098887) included additional mutations from other circulating Delta strains (R21T, T77K, E154K, Q216H, E482Q and H1099D) and retained the TM from SARS-CoV2 Spike but maintained the NDV/F CT, which was truncated to increase surface Spike. All chimeric constructs were subcloned into pCAGGS mammalian expression vector. RNA40D6 plasmid was generated by synthesizing the DNA fragment containing RNA40 sequence[28] between NDV leader and trailer sequences[56], and subcloned into the pCAGGS mammalian expression vector. An extra 3 nucleotides downstream of the NDV leader sequence were included to adjust the total number of NDV/RNA40 nucleotides divisible by 6.

VLPs were produced as previously described with some modifications[14]. The transfection was performed using the Expi 293 Expression System Kit [A14635] (Thermo Fisher Scientific, Waltham, MA, USA). Expi293F [A14527] (Thermo Fisher Scientific) cells were transfected using Expifectamine transfection reagent (Thermo Fisher Scientific) as recommended by the manufacturer. Briefly, 60 µg of plasmid mixture (For VLP H5: NP, M, HN and F/H5-vtn at the molar ratio of 1:1:1:1. For VLP-1086C-SOSIP or NFL-TD: NP, M, F/1086 C SOSIP or NLF-TD at the molar ratio of 1:1:1. For VLP CoV2: NP, M and F/CoV2 Delta+ at the molar ratio of 1:1:1 with 0.5 µg of TMPRSS2 plasmid) was transfected into $1.5 \times 108$ Expi 293 F cells. For all VLPs with TLR agonist incorporated, RNA40D6 plasmid was included in each plasmid panel at the same molar ratio. Expifectamine 293 transfection enhancers and 10 µg/mL of heparin were added to the culture 24 h post-transfection. Media containing VLPs were collected at 48 and 72 h post-transfection (72 and 96 h for VLP SOSIP or VLP NFL-TD) and purified by a series of discontinuous sucrose gradients, as previously described[14]. The media was centrifuged at $38,500 \times g$ for 18 h at 4 °C using a SW32 rotor in an Optima L-100K Ultracentrifuge (Beckman Coulter, Brea, CA, USA) to obtain a VLP pellet. The VLP pellet was then resuspended in TNE buffer (25 mM Tris-HCl, 150 mM NaCl and 5 mM EDTA, pH 7.4) and layered on top of a discontinuous sucrose gradient containing 20% and 65% sucrose (w/v) cushion, followed by centrifugation at $100,000 \times g$ for 6 h at 4 °C in a SW41 Ti rotor (Beckman Coulter). The VLP fraction collected between the 20% and 60% sucrose interphase was adjusted to 60% sucrose concentration and layered between a 50% and 80% sucrose gradient. The tube was then topped up with 10% sucrose and centrifuged at $200,000 \times g$ for 16 h at 4 °C. The interphase between 10% and 50% sucrose containing purified VLPs was collected, diluted in TNE buffer, and pelleted by centrifugation at $145,000 \times g$ for 6 h at 4 °C. The VLP pellet was resuspended in TNE buffer and stored at −20 °C until further use.

## Western blot
Cell lysates or purified VLPs were heat-denatured at 95 °C for 10 min in sample buffer under reducing conditions. Samples were resolved on 10% Tris-Glycine SDS-PAGE and transferred on a nitrocellulose membrane for Western blot analysis using the following antibodies: chicken anti-Newcastle Disease Virus polyclonal antibody [ab34402] (Abcam, Cambridge, United Kingdom) and goat anti-chicken IgY H&L- HRP conjugated secondary antibody [ab6877] (Abcam) for detection of Newcastle Disease Virus antigen; rabbit anti-HIV-1 gp120 Env (Clade B, IIB) antibody [ABL#5414] (Advanced Bioscience Laboratories, Rockville, MD, USA) and donkey anti-rabbit IgG (H + L) cross-absorbed- HRP conjugated antibody [SA1-200] (Thermo Fisher Scientific) for detection of HIV-1 Env protein; mouse anti-influenza A virus (H5N1/HA1) antibody [ab135382] (Abcam) and horse anti-mouse IgG-HRP conjugated antibody [7076S] (Cell Signaling Technology, Danvers, MA, USA) for detection of influenza HA protein; rabbit anti-spike (SARS-CoV2) antibody [scv2-SA-200] (eEnzyme LLC, Gaithersburg, MD, USA) and donkey anti-rabbit IgG (H + L) cross-absorbed-HRP conjugated

antibody [SA1-200] (Thermo Fisher Scientific) for detection of SARS-CoV-2 spike proteins. Signals were detected using the SuperSignal West Pico Plus Chemiluminescent substrate [34580] (Thermo Fisher Scientific) with the ChemiDoc MP Imaging system (Bio-Rad Laboratories) and analyzed by ImageJ (Version 2.1.0/1.53c).

## Flow cytometry
To examine the preservation of native-like Env conformation, HIV-1/NDV chimeric proteins were expressed in mammalian cell lines, and the binding of anti-Env antibodies was measured by flow cytometry. One day before transfection, $4.5 \times 10^6$ A549 human adenocarcinoma cells [CCL-185] (ATCC, Manassas, VA, USA) were seeded in T-75 flasks with F-12K medium [30–2004] (ATCC) containing 1% Penicillin-Streptomycin-Glutamine [10378016] (Thermo Fisher Scientific) and 10% Fetal Bovine Serum [100–106] (Gemini Bio-Products, West Sacramento, CA, USA). Cells were transfected with 15 µg of NDV-F/Env chimeric plasmid DNA, 75 µl of DNA-In® A549 Transfection Reagent [73772] (MTI-Global Stem, Gaithersburg, MD, USA), and 150 µl of Opti-MEM [31985070] (Thermo Fisher Scientific) and cultured for 48 h at 37 °C with 5% $CO_2$. To detect expression of Env, cells were collected 48 h post-transfection with 0.01 M EDTA in phosphate-buffered saline (PBS) and stained with 50 µl of anti-Env monoclonal antibodies PGT145, PG16, VRC01, b12, PGT151, 8ANC195, 35O22, 10E8, F105, or 447-52D (BEI Resources, Manassas, VA, USA) at 1 µg/ml in PBS containing 0.01 M HEPES and 0.09% bovine serum albumin [A7979] (Millipore Sigma, Burlington, MA, USA) for 1 hr at 37 °C. A secondary antibody, goat anti-human IgG Fab2-phycoerythrin (PE) [109-116-097] (Jackson ImmunoResearch, West Grove, PA, USA) was used at a 1:100 dilution for 1 hr at 37 °C. To differentiate live and dead cells, a Live/Dead Fixable Violet Dead Cell Stain Kit [L34964] (Thermo Fisher Scientific) was used at a 1:250 dilution for 30 min at room temperature. Cells were fixed with 250 µl of Cytofix/Cytoperm [554722] (Becton Dickinson, Franklin Lakes, NJ, USA) for 20 min on ice. Alternatively, A549 cells were infected with Ad4-Env at an MOI of 0.1, harvested at 48 h post-infection, and processed for surface staining. Cells were then permeabilized overnight in Perm/Wash buffer [554723] (Becton Dickinson) and intracellularly stained with 50 µl of anti-Hexon (adenoviral capsid protein) antibody 8C4-allophycocyanin (APC) [NB600-413APC] (Novus Biologicals, Centennial, CO, USA) at a 1:700 dilution in Perm/Wash buffer for 30 min on ice. Cells were analyzed by flow cytometry on a BD FACS Aria with FACSDiva software (Becton Dickinson).

To assess incorporation of viral glycoproteins, anti-Env antibodies, VRC01 or PGT145 were custom conjugated to PE (Becton Dickinson). VLPs were mixed with a diluted fluorescent primary anti-Env antibody in PBS containing bovine serum albumin (BSA) in 4.5-ml V-bottom polystyrene tubes and incubated for 30 min in the dark at 4 °C. After incubation, the sample was diluted 10× in PBS/BSA. VLPs were transferred to 5-ml polystyrene round-bottom tubes and were analyzed with a FACSymphony S6 cell sorter with FACSDiva software (version 10.9.0) (Becton Dickinson). VLPs not containing HIV-1 Env were stained with the fluorescent anti-Env antibodies to use as controls. The cytometer was set to trigger on both forward scattering (FSC) and side scattering (SSC) lights. VLPs were detected by FSC and SSC, and then the population of gated virions was determined to be expressing Env using fluorescence emitted from the anti-Env PE conjugated antibodies. To confirm that events were indeed VLPs, the FSC and SSC thresholds and voltages were adjusted to discriminate buffer particulates from VLPs using PBS/BSA without VLPs. Cleaning with BD Detergent Solution [660585] (Becton Dickinson) was performed as needed between each sample to ensure fewer than 50 events were detected in a tube of PBS/BSA collected over a minute at the maximum flow rate.

## Quantitative PCR (qPCR)
qPCR was performed using the QuantStudio 3 System (Thermo Fisher Scientific) to measure the levels of RNA40 incorporated into

VLP variants. RNA was extracted from VLPs by QIAamp Viral RNA Mini Kit [52904] (Qiagen, Venlo, Netherlands) with and without RNase treatment, then reverse transcribed by SuperScript III First-Strand Synthesis System [18080051] (Thermo Fisher Scientific) using random hexamers according to the manufacturers' instruction. Synthesized cDNA product was then combined with TaqMan Fast Advanced Master Mix [4444556] (Thermo Fisher Scientific). The RNA40D6 transcript was amplified using the following synthesized primers: forward primer 5'-CCAAAGAGTCGGAATTTAACGC-3', reverse primer 5'-TGTGAGGTACGATAAAAGGCG-3', and TaqMan probe labeled with a 5' reporter dye (FAM) and 3' fluorescent quencher (TAMRA dye): 5' (6-FAM)-ACGGAGTCACACAACAGACGGG-(TAMRA-Sp) 3'. The reaction conditions were as follows: one 20 s period at 95 °C, followed by 40 cycles of 1 s at 95 °C and 20 s at 60 °C. The Cq values were used to report the level of transcripts detected in copies/μg. Data were analyzed using the QuantStudio 3/5 Real-Time PCR software and Thermo Fisher Connect Platform (Thermo Fisher Scientific).

### Negative-stain electron microscopy
A 4.8-μl drop of the sample was applied to a freshly glow-discharged carbon-coated copper grid for ~15 s and removed using blotting paper. The grid was washed with three drops of buffer containing 0.01 M HEPES, 150 mM NaCl, pH 7.0, followed by negative staining with three drops of 0.75% uranyl formate. Micrographs were acquired using a Talos F200C transmission electron microscope (Thermo Fisher Scientific) operated at 200 kV and equipped with a Ceta CCD camera (Thermo Fisher Scientific). The nominal magnification was 57,000, corresponding to a pixel size of 2.53 Å. To estimate the number of visible spikes per VLP, micrographs were high-pass filtered to 250 Å using SPIDER[57] to suppress the signal corresponding to the VLP, followed by low-pass filtration to 15 Å to eliminate high-frequency noise. Spikes were then detected automatically in Relion 3.0[58] using a Laplacian-Gaussian filter with a minimal diameter of 90 Å and a maximal diameter of 150 Å. When the elongated shape of the spikes and their high density prevented reliable automatic quantification, visible spikes were counted manually instead.

### Rabbit immunization
Female NZW rabbits [Crl: KBL(NZW), stock number 052] (6–8 weeks old: 6 animals per group) (Charles River Laboratories) were immunized intramuscularly with 150–500 μg of purified VLPs in TNE buffer at 0, 4, 12, and 20 weeks and $10^{11}$ TCID$_{50}$ of Ad4 FED3 Env150 at 0 and 4 weeks for bolus immunization studies. One-fifth of a human dose of AS01 adjuvant (containing 10 μg of 3-O-desacyl-4'monophosphoryl lipid A (MPL) from *Salmonella minnesota* and 10 μg of QS-21, a saponin, combined in a liposomal formulation (GlaxoSmithKline Biologicals, London, United Kingdom) was formulated with VLPs for the groups tested to assess the impact of the adjuvant. For dose escalation immunization studies, a total of 500 μg of VLP was split into 7 doses (1, 2, 5.8, 15.8, 42.9, 116.5 and 316 μg) and administered intramuscularly in 48-h intervals. The adjuvant was also split proportionally to the VLP in these groups. The blood samples were collected at 0, 4, 8, 12, 14, 16, 20, 22, 24, and 28 weeks post-immunization.

### Purification of serum Ig
Serum Ig was purified using a 1:1 mix of rProtein A Sepharose™ Fast Flow and Protein G Sepharose™ 4 Fast Flow resins [17-1279-02 and 17061802] (Cytiva, Marlborough MA, USA) according to manufacturer's instructions. Briefly, 1 ml of rabbit serum from each animal was diluted (1:1) in Pierce™ Protein A/G IgG binding buffer [54200] (Thermo Fisher Scientific) and passed through a Poly-Prep® Chromatography gravity flow column [7311550] (Bio-Rad Laboratories, Hercules, CA, USA) packed with 500 μl of Protein A/G Sepharose. Columns were washed with PBS to remove non-specific binding and eluted with

IgG Elution Buffer (Thermo Fisher Scientific). Purified Ig was dialyzed in PBS and concentrated by centrifugation at $3000 \times g$ at 4 °C using a 30 kDa MWCO Amicon ultra centrifugal filter [UFC903008] (Millipore Sigma). Pierce™ BCA Protein Assay Kit [A55864] (Thermo Fisher Scientific) was used to quantify concentrated purified Ig and stored at −80 °C until use.

### Mesoscale discovery assay
HIV-1 Env-specific antibodies were assayed using the 384-well Streptavidin SECTOR Plate [L21SA-1] (Meso Scale Discovery, Rockville, MD, USA). To reduce non-specific binding signals, plates were blocked using a 5% MSD Blocker A solution [R93BAA] (Meso Scale Discovery) for 1 h with shaking using an Orbi-Shaker MP (Benchmark Scientific Inc., Edison, NJ, USA) at room temperature. Following blocking, plates were washed with 1× MSD Wash Buffer [R61TX-1] (Meso Scale Discovery) and incubated with the biotinylated capture protein with shaking for 1 h at room temperature. Plates were washed with 1× MSD Wash Buffer, and then serial dilutions of samples and controls were prepared in 1% MSD Blocker A [R93BA] (Meso Scale Discovery) in DPBS with 0.05% Tween-20, and then added to the plate and incubated for 1 h with shaking at room temperature. Samples were tested using serial dilutions starting at a minimum dilution of 1:100. After sample incubation, plates were washed with 1× MSD Wash Buffer, and 1 μg/mL goat anti-rabbit SULFO-TAG™ conjugated detection antibody [R32AB-1] (Meso Scale Discovery) was added for 1 h with shaking at room temperature. To detect signals 1× MSD Read Buffer was applied and analyzed using the MSD Sector Imager S600 (Meso Scale Discovery). All samples were tested in duplicate. Samples with a replicate coefficient of variation >30% were retested. Serial dilutions of the sample were used to assign an area under the curve (AUC) value as the primary readout. Results were plotted and analyzed using Prism version 9 or newer (GraphPad, San Diego, CA).

### Electron microscopy-based polyclonal epitope mapping
The details of serum and sample preparation to obtain polyclonal antigen-binding fragments (Fabs) for EMPEM were previously described[59]. Briefly, IgG was isolated from 1 mL rabbit sera (drawn at week 31 post-first immunization) using a self-packed FPLC column containing 5 mL CaptureSelect Fc multispecies resin [2942852005] (Thermo Fisher Scientific) on an AKTA Pure system (Cytiva). Polyclonal IgG was eluted with 0.1 M glycine pH 2.0, and buffer exchanged into TBS (50 mM Tris-HCl, 150 mM NaCl, pH 7.4). Papain [76216] (Millipore Sigma) was used to digest IgG to Fabs. Trimer-Fab complexes were prepared and incubated overnight by mixing 15 μg of 1086 C NFL, BG505 SOSIP or PVO.04 NFL trimers with 1 mg of Fab mixture (containing Fc and residual papain). The complexes were purified using a Superdex 200 Increase 10/300 GL gel filtration column [28990944] (Cytiva). Purified complexes were concentrated and diluted to a final concentration of 0.03 mg/mL, which were adsorbed on glow-discharged carbon-coated copper mesh grids and stained with 2% (w/v) uranyl formate. Electron microscopy images were collected on an FEI Tecnai Spirit T12 equipped with an FEI Eagle 4k × 4k CCD camera (120 keV, 2.06 Å/pixel) or a FEI Thermo Fisher Scientific Glacios equipped with a Thermo Fisher Scientific Falcon IV direct electron detector (200 keV, 1.89 Å/pixel) and processed using Relion 3.0[60] following standard 2D and 3D classification procedures. UCSF Chimera[61] was used to generate the composite maps and estimate epitope contacts by fitting the atomic coordinates of a BG505 SOSIP protomer into the map.

### Pseudovirus production and entry inhibition assay
HIV-1 Env, Flu-HA and SARS-CoV-2 S pseudoviruses were generated as previously described[44]. To produce HIV-1 Env pseudovirus,

HEK293T cells [CRL-3216] (ATCC) were co-transfected with plasmids encoding an Env-deficient backbone (pSG3ΔEnv) and HIV-1 Env (BG505, Q769.d22, KER 2018, SF162, BaL.01, PVO.04, Du156.12, ZM106.9, 16055, 1086 C) or SIV Env (SIVmac256) at a ratio of 3:1. For influenza virus HA pseudovirus, HEK293T cells (ATCC) were transfected with the following plasmids: 1 μg of HA, 0.1 μg of matched NA (A/Vietnam/1203/2004), 17 μg of pCMV-Luc and 0.1 μg of TMPRSS2. SARS-CoV2 pseudoviruses were generated by transfection of HEK293T/17 cells [CRL-11268] (ATCC) with the following plasmids: 0.53 μg of SARS-CoV-2 S (Wuhan, Delta+ S, Omicron B.1, or Omicron BA.2), 9.2 μg of lentiviral backbone (VRC5602), 9.2 μg of pCMV-Luc and 0.16 μg of TMPRSS2. Supernatants containing pseudoviruses were harvested 48-72 h post-transfection and processed by centrifugation at $2000 \times g$ for 10 min at room temperature and filtration through a 0.2 μm filter for titration. Processed pseudoviruses were titered using the following cell lines: TZM-bl [HRP-8129] (BEI Resources) for HIV-1, HEK293A [R70507] (Thermo Fisher Scientific) for Influenza, and HEK293T/ACE [631289] (Takara Bio Inc., Kusatsu, Shiga, Japan) for SARS-CoV2 and stored at −80 °C until further use.

HIV-1, Influenza and SARS-CoV-2 neutralization activities of sera from VLP-HIV-1/Flu/CoV2 immunized rabbits were tested using a single-round pseudovirus infection of TZM-bl, HEK293A or HEK293T/ACE cells, respectively, as described previously[23]. For testing HIV-1 and Influenza neutralization, heat-inactivated rabbit serum or purified serum Ig was serially diluted five-fold with Dulbecco's modified Eagle medium (Thermo Fisher Scientific) supplemented with 10% FCS (GeminiBio), and 10 μl of serum, purified Ig, or mAb was incubated with 40 μl of pseudovirus in a 96-well plate at 37 °C for 30 min. TZM-bl (for HIV-1), HEK293A (for Influenza) or HEK293T/17-Ace (for SARS-CoV-2) cells at $1 \times 10^4$ per well were then added, and plates were incubated at 37 °C with 5% $CO_2$ for 48 h (for HIV-1 and Influenza) or 72 h (for SARS-CoV-2). Luciferin signals were then detected using a Luciferase Assay System (Promega), and the relative light units (RLU) were read on a Victor X2 luminometer (Perkin Elmer, Waltham, MA, USA). All neutralization assays were performed in duplicate, and means are reported. All neutralization assays were repeated at least once with similar results.

To competitively inhibit V3-mediated SF162 pseudovirus neutralization, 10 μl of serially diluted heat-inactivated rabbit serum was co-incubated with 20 μl of solubilized V3 peptide or control peptide (EIYKRWII) at a concentration of 75 μg/μl for 90 min at 37 °C. Pseudovirus (20 μl) was then added before incubation at 37 °C for 30 min. TZM-bl cells ($1 \times 10^4$ cells/well) were then added to 96-well cell culture plates and incubated, as described above.

### Statistical methods
Welch's two-sample $t$ test (unequal variances) was used for all comparisons unless otherwise noted. ArUC was calculated by multiplying neutralizing magnitude at each timepoint by the number of timepoints using trapezoidal integration using R ve4.4.0. In cases where experiment length differed, the duration was shortened to match for comparison. To assess V3-mediated neutralization a paired $t$ test was used to compare control peptide to V3 peptide.

### Reporting summary
Further information on research design is available in the Nature Portfolio Reporting Summary linked to this article.

## Data availability
All data supporting the findings in this study are available within the article and its Supplementary Information. The source data underlying Figs. 1–7, Supplementary Figs. 1–6 and Supplementary Tables 1 and 2 are provided as a Source Data file. The negative-stain EMPEM map has been deposited in the Electron Microscopy Data Bank under accession code EMD-49656. Source data are provided with this paper.

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

## Acknowledgements

This research was supported in part by the Intramural Research Program and Vaccine Research Center of the National Institute of Allergy and Infectious Diseases, National Institutes of Health. It has also been funded in part with federal funds from the National Cancer Institute, National

Institutes of Health, under Contract No. 75N91019D00024. This research was also supported in part by the National Institute of Allergy and Infectious Diseases of the National Institutes of Health Award Number P01 AI157299 (A.B.W., R.W.). The content of this publication does not necessarily reflect the views or policies of the Department of Health and Human Services, nor does mention of trade names, commercial products, or organizations imply endorsement by the U.S. Government.

## Author contributions

K.M., P.D.K., R.W., and M. Connors led the design and execution of the study. K.M., M.H., E.W., J.P., A.A.P., L.B., B.K., T.G., S.S., D.R., I.S., A.P., T.S., N.E.W., J.D.W., F.V.V., R.R., E.K., P.M.R., E.O., and B.L. constructed, purified and characterized VLPs, constructed and purified adenovirus, and performed neutralization assays. Y.T. and T.S. performed electron microscopy and spike quantitation. I.D., M.S., I.B. and R.W.S. designed the VLP SOSIP constructs. L.W.M., M. Connors, and T.M. designed and managed the production of NDV VLPs. E.C. and J.L. performed statistics. J.L.T., R.N.L., A.S.T., G.O., and A.B.W. performed EMPEM analyses. G.D., L.S., S.N., B.L., and M. Castro performed antibody binding assays. J.G. and R.W. designed the VLP NFL-TD constructs. K.M. and M. Connors wrote the manuscript.

## Funding

## Competing interests

The authors declare no competing interests.

## Additional information

[1]HIV-Specific Immunity Section of the Laboratory of Immunoregulation, National Institute of Allergy and Infectious Diseases, National Institutes of Health, Bethesda, MD, USA. [2]Electron Microscopy Laboratory, Cancer Research Technology Program, Leidos Biomedical Research Inc., Frederick National Laboratory for Cancer Research, Frederick, MD, USA. [3]Department of Medical Microbiology and Infection Prevention, Amsterdam University Medical Centers, Location AMC, University of Amsterdam, Amsterdam, the Netherlands. [4]Department of Microbiology and Physiological Systems, Sherman Center, University of Massachusetts Medical School, Worcester, MA, USA. [5]Division Of Clinical Research, Biostatistics Research Branch, National Institute of Allergy and Infectious Diseases, National Institutes of Health, Bethesda, MD, USA. [6]Department of Integrative Structural and Computational Biology, The Scripps Research Institute, La Jolla, CA, USA. [7]Vaccine Research Center, National Institute of Allergy and Infectious Diseases, National Institutes of Health, Bethesda, MD, USA. [8]International AIDS Vaccine Initiative Neutralizing Antibody Center, Department of Immunology and Microbiology, The Scripps Research Institute, San Diego, CA, USA. ✉e-mail: mconnors@nih.gov

