## [Transparent Peer Review file · Nature Communications]

Antigen Persistence and TLR-Stimulation Contribute to Induction of a Durable HIV-1-Specific Neutralizing Antibody Response

Corresponding Author: Dr Mark Connors

Version 0:

Reviewer comments:

Reviewer #1

(Remarks to the Author)

The immunogenicity of HIV-1 Env glycoprotein is hindered by its structural instability and extensive glycan shielding, which is considered as the major barrier to develop an HIV-1 vaccine. In this manuscript, Kenta Matsuda et al. attempt to investigate factors that may influence the durable neutralizing antibody targeting HIV-1 Env by using a Newcastle Disease Virus-like particle (NDV-VLP) platform. The authors first display stabilized HIV-1 Env on the NDV-VLP, however, this strategy does not recapitulate similar neutralizing antibody response observed in influenza virus and SARS-CoV-2 models. Interestingly, a replicating Ad4 expressing HIV-1 Env could rapidly induce autologous neutralizing antibodies. The authors suppose that the activation of innate immunity and RNA packaging may improve the VLP system to induce Env-specific antibody responses. However, the overall neutralizing antibody response is very weak and this study does not provide any recommendations for HIV-1 vaccine design or strategy optimization. We still don't know how to induce the durable neutralizing antibody response against HIV-1.

1. Since Env displayed on VLP was barely able to induce HIV-1 neutralizing antibodies showed in Figure 3, it might be not appropriate to continue using VLP to investigate the immunization strategy.
2. The impact of activation of innate immunity in Line 191-227 didn't show notable changes and the impact of dose in Line 241-256 didn't achieve statistical significance. The impact of activating TLRs 4, 7 and 8 in combination with a high dose in Line 257 and 269 didn't show a significant improvement. These results are not enough to suggest the total amount of the antigen presented to the immune system has a significant impact on the magnitude of neutralizing antibody response.
3. Line 63-65: need more evidences to support "replicating viral vectors are often more immunogenic".
4. What is the difference between Figure 1B and 1C?
5. Please check the article for multiple miswriting about SARS-CoV-2, Omicron BA.1, and BA.2.

Reviewer #2

(Remarks to the Author)

The study "Antigen Dose and Persistence Contribute to Induction of a Durable HIV-1-Specific Neutralizing Antibody Response" by Matsuda et al. investigates the factors influencing the immunogenicity of stabilized HIV-1 Env glycoprotein and the durability of vaccine-induced humoral immune responses using a Newcastle Disease Virus-like Particle (NDV-VLP) platform. The immunogenicity of Env was compared to Influenza H5 hemagglutinin (HA) and SARS-CoV-2 Spike (S) proteins. Variables such as antigen conformation, dose, spike density, adjuvants and antigen persistence were analyzed. Key findings indicate that Env is significantly less immunogenic than HA and S, even when expressed at high valency and in different stabilized conformations (SOSIP and NFL-TD), highlighting its unique challenges in eliciting neutralizing antibodies (NAbs). While adjuvants modestly increased the magnitude of the antibody response, durability was primarily enhanced by high total doses and escalating dose schemes. However, these strategies still failed to achieve the immunogenicity observed with a replicating Ad4-Env vector, underscoring the need to identify key features of viral infection that could be leveraged for more effective vaccine design.

The study is well-structured, presenting a clear and logical progression from the research objectives to the conclusions. The methodology employed is novel and well-justified, demonstrating a rigorous and innovative approach to addressing the research question. The results are robust and provide support for the proposed hypotheses, offering valuable insights that

contribute significantly to the field. The authors contextualize their findings by comparing them with previous research not only in HIV vaccine development, but also in the broader context of long-lasting immune responses against other pathogens and the relevance of replicating vaccines. However, the study lacks detailed comparisons with recent approaches using other VLP systems or nanoparticle-based Env presentations, which could further contextualize its findings within the latest advancements in the field.

Given the well-documented challenges in HIV-1 vaccine development, this study reinforces the idea that a long-lasting neutralizing antibody response can be achieved by combining multiple factors—Env stabilization, valency, innate immune stimulation, total dose, and antigen persistence. Addressing these aspects in future studies could provide deeper insights and further refine strategies for developing more effective HIV vaccines. Overall, this work represents an important advancement, with potential implications for further research and practical applications.

Minor concern:

- The study provides a thorough assessment of Env valency on the VLP surface using negative stain electron microscopy; however, it is unclear whether similar valency estimation was performed for the Influenza H5 and SARS-CoV-2 S proteins. Given the importance of valency in antigen presentation and immune activation, could the authors clarify how the valency of S and H proteins compares to that of Env? Additionally, since the spatial arrangement of surface spikes can influence antigen recognition and B cell activation, it would be valuable to understand how the spacing of Env trimers on NDV-VLPs compares to that on nanoparticle-based platforms expressing stabilized trimers. Could the authors clarify whether variations in valency or antigen distribution between the two stabilization approaches might impact key immunological outcomes, such as neutralizing antibody titers or B cell activation?

- The assessment of Env incorporation in HIV-NDV VLPs presents some uncertainties. In its natural form, HIV Env consists of gp120 and gp41, which are non-covalently bound, making their interaction relatively unstable. This instability raises the possibility that ultracentrifugation steps during VLP purification could lead to partial dissociation and loss of gp120 Env proteins. In a Western blot using an anti-gp120 antibody on cell lysates from transfected cells, the non-stabilized HIV-1 Env (F1086C) band appears as intense as that of the stabilized variants. However, in the Western blot of purified VLPs, the signal for this band is significantly reduced. Therefore, the conclusion that F/1086C exhibits lower Env incorporation into the particles (lines 110–112) requires further validation to rule out potential losses introduced by ultracentrifugation.

- In Figure 3F, explicitly indicating the values for the SIV (negative control). Including these values would help assess the non-specific response.

- The interpretation of the breadth of NAbs in immunized animals should be reconsidered in light of the negative control data. Specifically, in lines 299–300, the statement that some animals neutralized all tested strains, including tier 2 and 3 isolates, does not account for the magnitude of NAbs detected in the control group. Could the authors clarify this aspect and consider revising the conclusions accordingly?

- Could the authors discuss whether additional modifications to the stabilization strategy, such as glycan shielding adjustments, trimerization domain refinements, or immune-enhancing scaffolds, could further improve Env immunogenicity? Additionally, do the authors consider that expressing these stabilized Env constructs from the replicating adenoviral vector could yield better immunogenicity outcomes compared to the non-stabilized version used in the study for comparative purposes?

Reviewer #3

(Remarks to the Author)

Matsuda and colleagues present a clear and concise manuscript describing evaluation of elegantly conceived and well-executed novel approaches to induction of anti-HIV-1-Env neutralizing antibodies. They present strong evidence that in their rabbit experiments, numerous features of infection or vaccines determine neutralizing antibody and appropriately suggest that these might be useful for improvement of HIV-1 vaccine immunogenicity.

The key results are 1) using stabilizing mutations in a high-valency NDP-VLP platform induces neutralizing antibodies for influenza H5 HA and SARS-CoV-2 S but not for HIV-1 Env, unlike Ad4 replication competent immunization, which does induce anti-Env neutralizing responses. Regarding anti-Env neutralization, 2) VLP vaccination with RNA40 RNA encapsidation or AS01 did not significantly increase anti-Env neutralization, but higher administered dose with RNA40 did improve responses, with an incremental benefit when administered with AS01 adjuvant, and 3) an escalating multi-dose strategy yielded higher magnitude and breadth of responses.

The authors conclude that their experiments demonstrate that poor neutralizing immunogenicity from Env observed in non-replicating vaccine concepts including their VLPs can be overcome by replicating virus vaccines, and that factors similar to replicating virus infection including increasing dose, TLR stimulation, and persistence of antigen, may be approaches that might be used to improve immunogenicity in non-replicating vaccine constructs. The discussion is thoughtful and does not overstate conclusions.

The manuscript includes appropriate construct characterization including associated figures 1 and 2 that establish construct validity and the immunogenicity experiments follow established methods, and methods are sufficient to allow for future replication by experienced researchers. Although the focus of the paper is on enhancing VLP immunogenicity against HIV-1

Env, inclusion of influenza and SARS-Cov-2 constructs provides important controls, as does including data using a replication-competent Ad4 HIV-1 vaccine construct. The figures are clear, and I agree with the author designation of some figures as supplemental.

The findings presented in this manuscript address an extremely important challenge to creation of an effective HIV-1 vaccine, which remains elusive despite massive effort over multiple decades. I would expect insights contained in this publication to form the bases for testable modifications to existing and novel vaccine constructs designed to elicit Nabs in humans, in whom immune responses may differ from those seen in rabbits.

I have only a few minor suggested improvements for consideration. First, I would revisit the title of the paper, which currently focuses on antigen dose and persistence, in contrast with the abstract conclusion, which properly summarizes the conclusions to include a broader list of features including dose, persistence, stabilization, spike density, and TLR stimulation. Also, I would consider slight modification of the narrative surrounding replicating vectors for HIV-1 preventive vaccination in lines 429-433. Although the statements there are correct, I would opine that the data presented in this manuscript indeed point the way toward non-replicating approaches, but the data also reinforce the immunologic superiority (hopefully just for now...) of replicating approaches, perhaps providing some rationale to follow that path and address safety or transmission concerns.

Following prompting contained in instructions to authors: I am a clinician with HIV-1 vaccine development primarily in human safety and immunogenicity trials.

Version 1:

Reviewer comments:

Reviewer #1

(Remarks to the Author)

My concerns have been addressed.

Reviewer #2

(Remarks to the Author)

The authors have thoroughly addressed all the concerns and suggestions raised in the previous round of review. The revised manuscript incorporates all the necessary clarifications and improvements.

I am satisfied with the current version and support its publication in its present form.

Reviewer #3

(Remarks to the Author)

The rebuttal statements and amended materials satisfactorily address my questions and suggested edits.

Reviewer #1 (Remarks to the Author):

- 1. Since Env displayed on VLP was barely able to induce HIV-1 neutralizing antibodies showed in Figure 3, it might be not appropriate to continue using VLP to investigate the immunization strategy.**

We thank the reviewer for their time and effort in reviewing this manuscript. We agree that if one's focus is to empirically induce neutralizing antibodies, our data support the idea that using VLPs as they are currently formulated is not a promising path. For this reason, viral vectors are a large part of our pre-clinical work and clinical trials. However, the VLP platform is highly immunogenic for RSV, Flu H5, and SARS-CoV-2. We are left with the clear result that, unlike a VLP, a replicating viral vector expressing the same antigen does rapidly induce HIV neutralizing antibodies. There is some immunology here that must explain the difference and understanding that in a variety of non-replicating platforms might advance future immunization strategies.

2. The impact of activation of innate immunity in Line 191-227 didn't show notable changes and the impact of dose in Line 241-256 didn't t achieve statistical significance. The impact of activating TLRs 4, 7 and 8 in combination with a high dose in Line 257 and 269 didn't show a significant improvement. We now make these statistical comparisons. The issue here is that we are making incremental improvements that aren't significant individually but are very different outcomes from the starting condition. For example, the data in figures 5 A,C, and D look quite different from Figure 1 C and D. We now include Supplemental Table 1 that shows the p value for all comparisons including the starting condition that is 150 µg. Consistent with the appearance of the change in the figures, the combination of high dose and TLR stimulation does make a highly significant difference now reported on lines 318-322.

3. Line 63-65: need more evidences to support “replicating viral vectors are often more immunogenic”. We have now added additional references to bolster the point on lines 81-82 of the revised manuscript.

4. What is the difference between Figure 1B and 1C? Figure 1B shows the Western of transfected cell lysates to show expression. Figure 1C shows westerns of vlp lysates and shows incorporation into VLPs. We have added labels to make the difference more obvious.

5. Please check the article for multiple miswriting about SARS-CoV-2, Omicron BA.1, and BA.2. We have now corrected the references to Omicron to make them uniform.

Reviewer #2 (Remarks to the Author):

- The study provides a thorough assessment of Env valency on the VLP surface using negative stain electron microscopy; however, it is unclear whether similar valency estimation was performed for the Influenza H5 and SARS-CoV-2 S proteins. Given the importance of valency in antigen presentation and immune activation, could the authors clarify how the valency of S and H proteins compares to that of Env? We thank the reviewer for their time and effort in reviewing this manuscript. We now include the valency of these on lines 145-146 and a statement of whether this might be the cause of differential responses on lines 450-452.

Additionally, since the spatial arrangement of surface spikes can influence antigen recognition and B cell activation, it would be valuable to understand how the spacing of Env trimers on NDV-VLPs compares to that on nanoparticle-based platforms expressing stabilized trimers. Could the authors clarify whether variations in valency or antigen distribution between the two stabilization approaches might impact key immunological outcomes, such as neutralizing antibody titers or B cell activation? We have now measured the spacing and included a statement on lines 147-149, and the results are discussed on lines 452-455.

- The assessment of Env incorporation in HIV-NDV VLPs presents some uncertainties. In its natural form, HIV Env consists of gp120 and gp41, which are non-covalently bound, making their interaction relatively unstable. This instability raises the possibility that ultracentrifugation steps during VLP purification could lead to partial dissociation and loss of gp120 Env proteins. In a Western blot using an anti-gp120 antibody on cell lysates from transfected cells, the non-stabilized HIV-1 Env (F1086C) band appears as intense as that of the stabilized variants. However, in the Western blot of purified VLPs, the signal for this band is significantly reduced. Therefore, the conclusion that F/1086C exhibits lower Env incorporation into the particles (lines 110–112) requires further validation to rule out potential losses introduced by ultracentrifugation. The reviewer correctly interprets the result. The result was validated by EM. We now emphasize that the loss of 120 on VLP during purification is corroborated by the EM result on line 147.

- In Figure 3F, explicitly indicating the values for the SIV (negative control). Including these values would help assess the non-specific response. We now include these values.

- The interpretation of the breadth of NABs in immunized animals should be reconsidered in light of the negative control data. Specifically, in lines 299–300, the statement that some animals neutralized all tested strains, including tier 2 and 3 isolates, does not account for the magnitude of NABs detected in the control group. Could the authors clarify this aspect and consider revising the conclusions accordingly? We do conclude that this additional breadth is nonspecific

on line 375. We do go on to characterize this nonspecific response at some length including additional specificity controls (lines 382-398) and EMPER (lines 413-436). We also emphasize that this low-level nonspecific antibody is unlikely to contribute to protective efficacy (lines 514-518).

- Could the authors discuss whether additional modifications to the stabilization strategy, such as glycan shielding adjustments, trimerization domain refinements, or immune-enhancing scaffolds, could further improve Env immunogenicity? Additionally, do the authors consider that expressing these stabilized Env constructs from the replicating adenoviral vector could yield better immunogenicity outcomes compared to the non-stabilized version used in the study for comparative purposes? We now at least mention these concepts in the discussion on lines 537-538. This is a topic of a very large manuscript in preparation, but beyond the scope of the current one. Comparing the Ad4 platform with stabilized and non-stabilized Env forms is also the subject of our clinical trial here at the Clinical Center.

Reviewer #3 (Remarks to the Author):

I have only a few minor suggested improvements for consideration. First, I would revisit the title of the paper, which currently focuses on antigen dose and persistence, in contrast with the abstract conclusion, which properly summarizes the conclusions to include a broader list of features including dose, persistence, stabilization, spike density, and TLR stimulation. We thank the reviewer for their time and effort in reviewing this manuscript. We have now added TLR stimulation to the title. We do not have data that varies spike density over a continuum so we would prefer to leave that to the reader.

Also, I would consider slight modification of the narrative surrounding replicating vectors for HIV-1 preventive vaccination in lines 429-433. Although the statements there are correct, I would opine that the data presented in this manuscript indeed point the way toward non-replicating approaches, but the data also reinforce the immunologic superiority (hopefully just for now...) of replicating approaches, perhaps providing some rationale to follow that path and address safety or transmission concerns. We have now modified the text to put the possible paths forward into context per the reviewer's comment on lines 533-535. We certainly agree with the reviewer, evidenced by our actions. We have moved replicating vectors into larger animals and human trials while we continue to try to understand the factors that might improve immunogenicity of non-replicating approaches.